# Design and Implementation of a Futuristic EV Energy Trading System (FEETS) Connected with Buildings, PV, and ESS for a Carbon-Neutral Society

Sangmin Park [1] , SeolAh Park [2] , Sang-Pil Yun [1,3], Kyungeun Lee [1,3], Byeongkwan Kang [1] , Myeong-in Choi [1] , Hyeonwoo Jang [4] and Sehyun Park [1,4,*]

1  Department of Intelligent Energy and Industry, Chung-Ang University, Seoul 06974, Republic of Korea; motlover@cau.ac.kr (S.P.); sethyun0215@cau.ac.kr (S.-P.Y.); kyungeun20@cau.ac.kr (K.L.); byeongkwan@cau.ac.kr (B.K.); auddlscjswo@cau.ac.kr (M.-i.C.)
2  Department of Industrial Security, Chung-Ang University, Seoul 06974, Republic of Korea; lwpark@cau.ac.kr
3  Korea Hydro & Nuclear Power Co., Ltd. (KHNP), Gyeongju-si 38120, Republic of Korea
4  School of Electrical and Electronics Engineering, Chung-Ang University, Seoul 06974, Republic of Korea; gostub123@cau.ac.kr
*  Correspondence: shpark@cau.ac.kr; Tel.: +82-2-822-5338

**Abstract:** To realize carbon neutrality, understanding the energy consumed in the building sector, which is more than that in other sectors, such as industry, agriculture, and commerce, is pivotal. Approximately 37% of energy consumption belongs to the building sector, and management of building energy is a critical factor. In this paper, we present an energy sharing scenario for energy stabilization, assuming that electric vehicles and their charging stations are widely distributed in the future. Consequently, fewer fuel cars will exist, and electric cars will become the major mode of transportation. Therefore, it is essential to install charging stations for electric vehicles in the parking lots of future buildings, and business models are expected to expand. In this paper, we introduce a future energy stabilization mechanism for peak power management in buildings and present a platform that entails connection-based energy trading technology based on a scenario. We also propose an energy supply strategy to prevent excess prices incurred due to peak consumption. Then, we analyzed the electricity bill for one month through scenario-based simulations of an existing building and the proposed system. When applying the proposed system, we derived a result that can reduce electricity rates by 38.3% (best case) to 78.5% (worst case) compared with the existing rates.

**Keywords:** ICT carbon-neutral platform; ICT carbon-neutral city modeling; zero-carbon city; urban energy analysis; new energy industry; smart energy city



## 1. Introduction

The world is rapidly changing due to the advent of the fourth industrial revolution and hyper-connectivity based on advanced information and communications technology (ICT) [1]. The proportion of traditional energy sources such as coal and nuclear power is decreasing, whereas the proportion of new and renewable facilities and power generation that includes solar power is increasing, and distributed energy resources such as energy storage systems (ESSs) and electric vehicles (EVs) are also expanding [2]. In addition, the electricity market has recently formed a trading market with a horizontal structure in which many operators and consumers trade electricity in a simple vertical structure. In the case of Korea, according to a research report by the Korea Institute of Construction Technology (KICT), the forecast for greenhouse gas emissions in the domestic building sector was 156.8 million tons in 2020, increasing to 173.3 million tons in 2025 and 189 million tons in 2030. Furthermore, the net emission from the building sector in 2018 was 52.1 million tons, which will continue to increase [3].

In the '2030 National Greenhouse Gas Reduction Goal (NDC)', the building sector aimed to reduce emissions by 32.8% compared with 2018 (52.1 million tons), achieving 35 million tons by 2030 [4]. In addition, the '2050 Carbon Neutral Scenario Plan' aimed to reduce emissions by 88% to achieve the goal of 6.2 million tons [5]. Aligning with this goal, the Ministry of Land, Infrastructure and Transport has established a roadmap for 2050 carbon neutrality in the building sector. To this end, it announced plans to achieve zero energy in new buildings and to expand green remodeling of existing buildings.

An announcement from the Secretary General of the UN Environment Program stated that to reduce carbon dioxide emissions from the building and construction sectors, three strategies are needed: (1) a significant reduction in energy demand in the built environment, (2) decarbonization of the power sector, and (3) the use of materials that can reduce carbon cycle emissions. The statement also emphasized that the transition of the building and construction sector to a low-carbon power generation path will greatly benefit economic recovery, making this transition a top priority for governments. According to the International Energy Agency (IEA), to achieve carbon neutrality in the building sector by 2050, direct carbon dioxide emissions from buildings must be reduced by 50% by 2030 and by 60% in the indirect sector.

### 1.1. The Necessity of Platform-Based Integrated Energy Management

There are methods to stabilize energy in a building and reduce energy peaks through analysis of building energy demand and supply. For example, consider that building B has an energy IoT and photovoltaic (PV) system. It will collect energy data through IoT and store the data in a server. Pattern analysis of historical energy data makes it possible to predict future demand-side energy consumption. Renewable energy is limited by various environmental factors such as climate and weather, and there are many extremely unstable factors, so the supply forecasting technology is an optimization method for optimizing the unstable energy [6]. Through supply forecasting, it is possible to predict how much energy can be supplied to demand [7].

Until now, the stabilization of building energy has been solved within an independent domain within a building. In other words, as shown in Figure 1, the building supplied energy independently from PV–ESS, and if necessary, energy was supplied through the energy transactions within the same domain, such as B2B (Building to Building) and H2H (Home to Home). However, since renewable energy is limited by various environmental factors such as climate and weather, and there are many extremely unstable factors, an independent domain is insufficient to supply energy to buildings stably. For a carbon-neutral society, we must consider energy trading and sharing mechanisms through linkages between small platforms (different domains), rather than such independent and fragmented energy sharing. In order to achieve a carbon-neutral society, we introduce a carbon-neutral platform in Section 3 and suggest the necessity of integrated energy management through linkage with other small platforms.

### 1.2. The Concept of Used Energy Trading Platform and Prosumer

Currently, the second-hand trading market is active in Korea. It reflects the sales psychology of consumers who wish to buy products at low prices or sell products at an optimized price. In the future energy field, an energy trading platform can be formed for the proper benefit of consumers by selling surplus energy through the Home to EV (H2E), Building to EV (B2E) based energy trading platform [8]. When the energy trading platform is activated, it is expected to become an innovative service as a platform for buying and selling surplus energy among consumers in the future as a smart energy prosumer platform [9]. The key point is that an EV to ESS system can be a pivotal technology for building energy stabilization. In other words, the battery installed in an electric vehicle is used as an energy storage system (ESS) or energy carrier. High-capacity batteries installed in electric vehicles can perform roles such as power supply in emergency situations and stabilization of power supply and demand. For example, based on the battery capacity

(72.6 KWh) of Korea EV Company A, the electricity can be used for about 10 days at home (based on average daily power consumption (7.3 KWh) per household in Seoul). Therefore, although EVs are currently applied only in the transportation field, in the future, batteries embedded in EVs will have an important role in energy sharing.

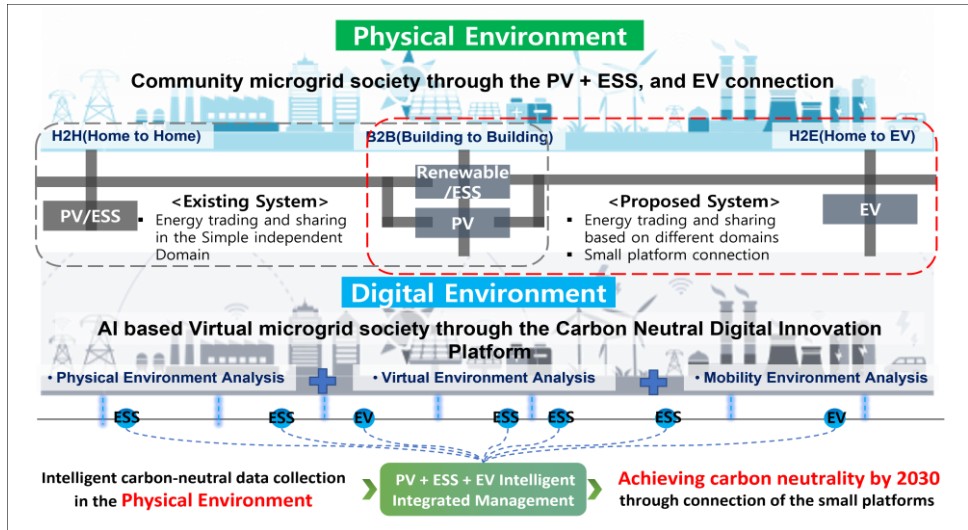

**Figure 1.** Concept of Proposed PV + ESS, and FEETS Connection System.

Figure 1 shows the concept of proposed PV + ESS, and FEETS Connection System. The physical environment represents an actual physical domain, and the digital environment represents a virtual domain for optimal data-based energy management. To achieve the complete carbon neutrality of the city, it is important to share energy in the local domain, community domain, and town domain through linkage with small platforms.

This paper proposes an energy-efficient building model that can be an effective alternative for carbon emission reduction in the building sector for carbon reduction. In addition, this paper utilizes the Internet of Things (IoT) to introduce a distributed-IoT sensor constructed in the building and explains the mechanism of demand forecasting in the building and supply forecasting in the photovoltaic (PV) system. As mentioned above, analysis of the building demand and distributed power supply is crucial for energy stabilization and peak power management in buildings. Renewable energy is highly influenced by weather and the environment, and thus has irregular characteristics. However, if the energy supply does not work smoothly through the PV/ESS platform due to the influence of the weather, another energy source must be identified. We introduce an expanded energy trading system based on B2E (Building to EV) and H2E (Home to EV) concepts as well as B2B and H2H, and link data through a carbon-neutral digital innovation platform for optimal management. Therefore, this paper proposes an alternative for stabilizing building energy and preventing peak power through an efficient energy supply in the building energy management system (BEMS), PV/ESS, and EV/vehicle-to-grid (V2G) as B2E and H2E. Table 1 shows the difference between the existing system and proposed system.

Therefore, the key elements of the proposed paper can be summarized as follows.

- In Section 3, we introduce the Carbon-Neutral Digital Innovation Platform and present the importance of energy strategy through the small platform connection method.
- We design a futuristic energy trading business model and accompanying PV and EV charging and trading platform based on an energy sharing mechanism in Section 5.
- We introduce Korea's energy policy and contract power and propose an energy supply strategy based on the energy supply and demand mechanism to respond to contract power in Section 6.
- We analyze the electricity bill for one month through scenario-based simulations of an existing building and the proposed system in Section 6.

**Table 1.** Difference of Existing System and Proposed System.

| Class. | Existing System | | Proposed System |
|---|---|---|---|
| Service domain | - | Simple independent domain connection (B2B, H2H, independent type) | - Small platform connection based multiple domains (B2E, H2E, etc., expandable type) |
| Discrimination in services | - | it does not exist; future applications for future energy trading | - User App-based futuristic energy trading system to prepare the increase in EV in the future |
| Service application area | - | Small domain such as a building, home, etc. | - City-based community, town area, etc. |
| DR management | - | Independent management of energy demand and supply | - Optimized energy demand and supply management with 3 step-based real-time energy monitoring |
| Energy peak control | - | Energy peak control is not properly performed due to unstable characteristic of renewable energy | - Optimized peak control through multi-platform connection for zero energy and economic benefit |

## 2. Related Work

Recently, there were several studies on systems for carbon neutrality and energy saving by connecting with PV, ESS, and EV Systems. We closely explored related works based on three factors: (1) service, (2) platform, and (3) physical infrastructure. Figure 2 shows the related work domain analysis and difference of the proposed system.

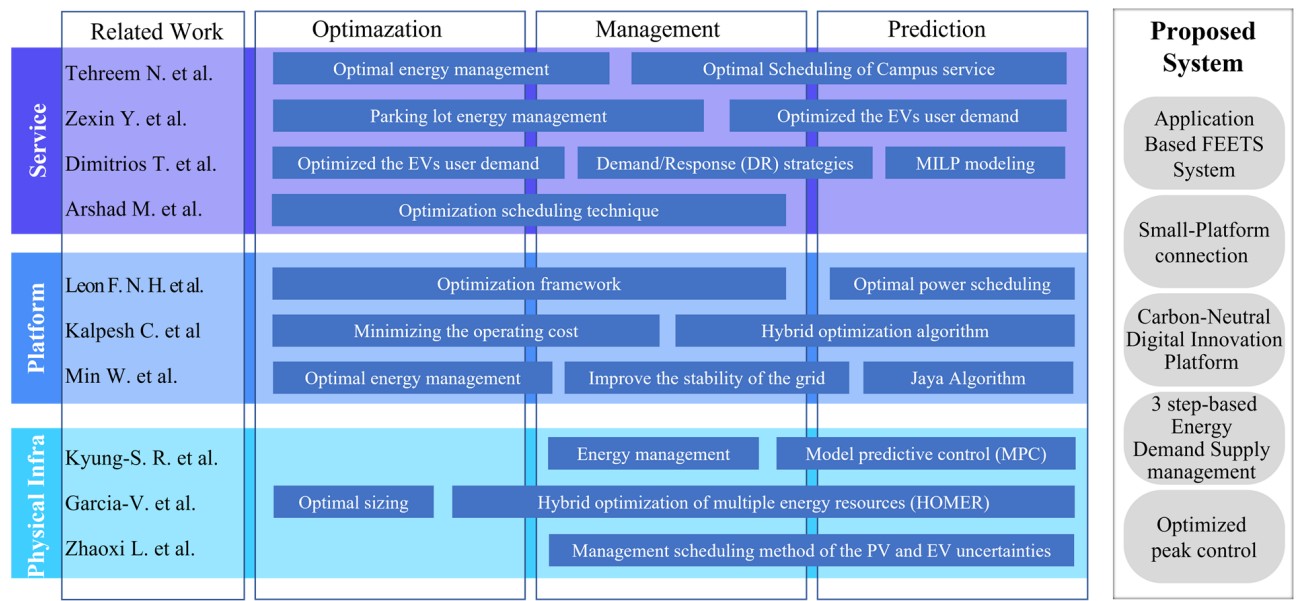

**Figure 2.** Related Work Domain Analysis and Difference of Proposed System [10–19].

### 2.1. Service Aspect

Tehreem N. et al. proposed the integration of photovoltaic (PV) systems, energy storage systems (ESS), and electric vehicles (EVs) on university campuses [10]. Additionally, the authors proposed an optimal energy management system (EMS) to optimally distribute energy from available energy sources. The simulation results show that the proposed EMS ensures continuous power supply and reduces energy consumption costs by nearly 45%. In addition, energy was saved by 45.58% using the EVs as energy sources. Zexin Y. et al. proposed a parking lot energy management system that integrates an energy storage system (ESS) and a photovoltaic (PV) system [11]. The authors introduced the concept of Energy

Price Tag (EPT) to define the price of any energy storage device and establish priority among PV, ESS, EV, and grid. In addition, this study optimized charging plans for ESS, EV, and buildings by considering EV user demand and PV power, with the optimization goal of minimizing charging cost. In the work of Dimitrios T. et al., a renewable energy source (RES), EV, and ESS integration strategy was demonstrated to provide additional energy and storage options to a microgrid [12]. The evolution of the smart grid enables end-users to actively participate in energy management systems (EMSs) through demand response (DR) strategies. The study evaluated the two-way energy trading capacity of EVs according to stochastic EV operation schedules using mixed integer linear programming (MILP) modeling and analyzed the influence of uncertainty on PV power generation. The authors studied how efficiently a BEMS operates, considering reducing costs by prioritizing different power sales. Arshad M. et al. illustrated the home-to-grid (H2G) system through EV, ESS, and PV integration [13]. The authors presented a plan for efficiency and an optimization scheduling technique in terms of energy and cost considering the real-time pricing (RTP) cost.

### 2.2. Platform Aspect

Leon Fidele Nishimwe H. et al. proposed an optimization framework for profit maximization that determines the integrated plan and operation of charging stations by considering vehicle arrival patterns, intermittent solar power generation, and energy storage device management [14]. This optimization framework finds the optimal configuration of grid-connected charging stations and determines the optimal power scheduling at the charging stations during operation. Through this platform, it performs the optimal energy saving management of electric vehicles, charging stations, and PV power generation. Kalpesh C. et al. presented an electric vehicle charging station and a hybrid optimization algorithm for energy storage management [15]. This algorithm consists of three parts: classification of real-time electricity rates by price range, real-time calculation of PV power based on solar insolation data, and optimization to minimize the operating costs of electric vehicle charging stations combined with PV and ESS. To confirm the efficiency of this algorithm, an extensive simulation study was conducted using a statistical EV charging model in the context of Singapore. Min W. et al. presented optimal energy management based on the Jaya algorithm for energy flow control in a smart home that includes solar power generation (PV) integrated with ESS and EV [16]. The Jaya algorithm controls home-to-vehicle (H2V) and vehicle-to-home (V2H) modes, buying energy from the grid and selling surplus energy to the grid.

### 2.3. Physical Infra. Aspect

Kyung-Sang R. et al. presented improving the hosting capacity of solar photovoltaic (PV) with energy storage systems (ESS) and electric vehicles (EV) in off-grid-based microgrids considering a predictive control (MPC) energy model [17]. Garcia-Vazquez C. A. et al. presented methods on how to improve energy management by integrating the PV, wind turbine (WT), and ESS in a home into a hybrid renewable energy system, determining optimal sizing through hybrid optimization of multiple energy resources (HOMER) and using vehicle-to-home (V2H) appropriately [18]. Zhaoxi L. et al. proposed a real-time EV charging management technique for building energy management systems (BEMS) in commercial buildings that provide solar power generation and EV charging services [19]. Through the suggested method, BEMS can maximize economic feasibility by managing PV and EV uncertainties and scheduling the grid exchange.

## 3. System Configuration

### 3.1. System Architecture

In this section, we introduce the Carbon-Neutral Digital Innovation Platform (CNDIP) and its role and types. In addition, the proposed system architecture is presented in connection with this CNDIP.

### 3.1.1. Carbon-Neutral Digital Innovation Platform (CNDIP)

Figure 3 shows the five-layer-based CNDIP architecture of the proposed system [20]. In addition, it shows the platform connection for BEMS, PV/ESS, and EV/V2G. In the overall flow, the infrastructure layer shows the sensors deployed in the building. The environmental information sensor consists of a temperature/humidity sensor, a $CO_2$ sensor, an occupancy sensor, and a power sensor, and the data collected from each sensor are primarily collected through the gateway. The temperature/humidity, $CO_2$, occupancy, and power sensors together with the gateway can be considered one IoT package infrastructure [21]. In this way, these data are transmitted to the central server through the gateway, and the data are stored in the database (DB). In addition, among the infrastructure layers, the PV + ESS domain also belongs to the infrastructure layer, and the energy collected from PV generation is stored as the ESS. Here, power generation and charging and discharging data are collected from the IoT and installed in the PV system and ESS. The IoT also collects data about EVs. The DB from which data are collected can be viewed as a big-data layer. This structure performs data analysis in the digital layer through the collected big data, and services are provisioned in the service layer by performing data analysis and prediction [22,23]. A description of each layer is shown below.

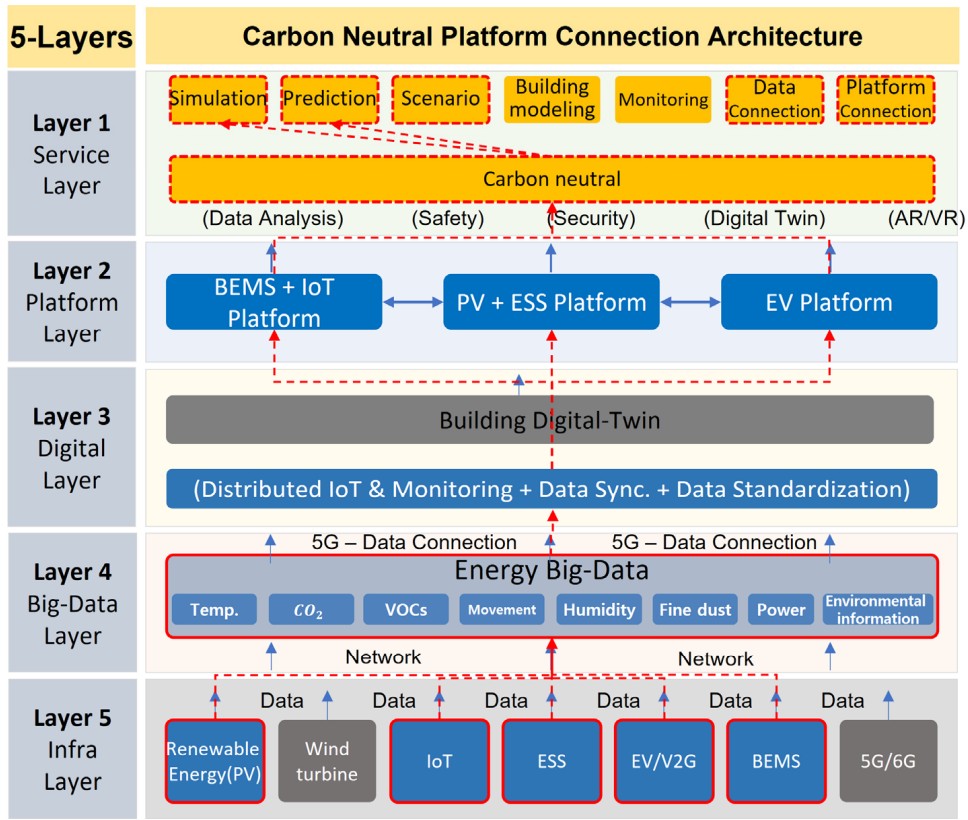

**Figure 3.** Carbon-neutral digital innovation platform (CNDIP).

- Infrastructure layer: The most important element in the infrastructure layer is the energy IoT. The energy IoT refers to a service that maximizes the energy efficiency through energy information collection, load management of the energy demand, and energy sharing/transactions by developing IoT-based smart energy platform technology to solve energy problems in a hyper-connected society. In addition, data are one of the important elements in the infrastructure layer. Representatively, there are various types of data (temperature, humidity, movement, power, etc.) from BEMS smart sensors and advanced metering infrastructure (AMI), renewable energy generation/facilities, fuel cells, PV infrastructure, and wind power generation infrastructure.

- Big-data layer: The big-data layer manages carbon-neutral data and analyzes energy data to optimize various types of carbon-neutral data from the infrastructure layer.
- Digital layer: The digital layer is responsible for data analysis and intelligence processing to create meaningful values of big data from the infrastructure layer. In addition, it is possible to link digital twins through a federated digital twin.
- Platform layer: Creating new value through platform linkage is becoming an essential technology. Through the connection between the previously built platform and the newly created platform, it is possible to build a customized platform for users. Therefore, the platform layer has a role in connecting various platforms of carbon neutrality and energy.
- Service layer: In the service layer, analysis prediction can be performed through data received from the infrastructure layer, and a business model can be built as a service element.

### 3.1.2. Proposed System Architecture (Connection of CNDIP)

Figure 4 shows the architecture of the proposed system connected with the CNDIP platform. The architecture is connected to three domains (building, PV/ESS, and EV) and provides smart services and business models.

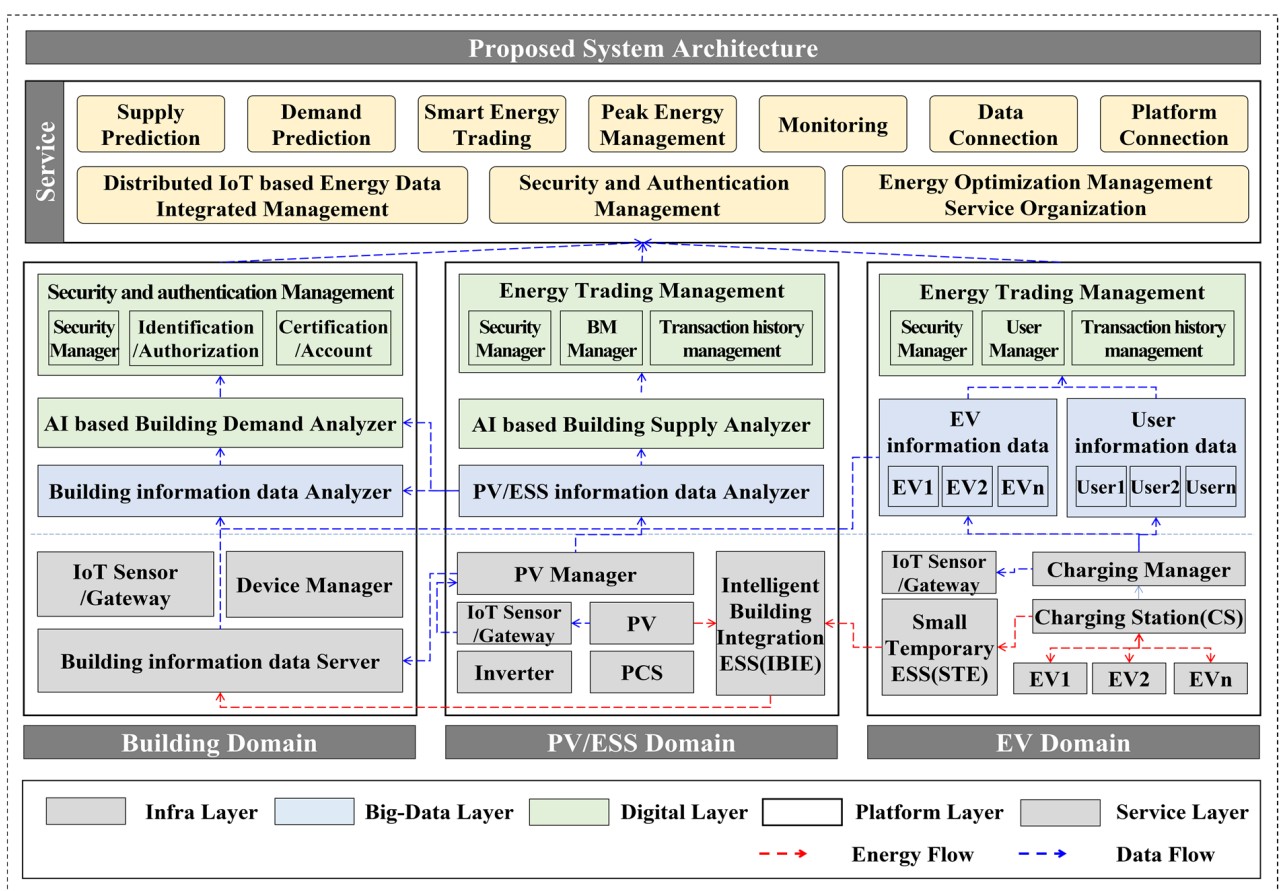

**Figure 4.** Proposed System Architecture.

Table 2 shows a detailed description of the proposed architecture and describes related services and layers. Each building, PV/ESS, and EV domain is composed of modules, and each module has the function of the contents of Table 2. In the infra layer, the IoT sensors gather the energy demand environment information and send the data to the server. There are many types of data such as building management data, user location data, device and sensor data, and user behavior data in the building information data server [24]. The Digital layer classifies building information data, PV/ESS information data, user/EV information

data, and real time energy monitoring analysis data coming from the Big-data layer for the demand/supply forecast and management service [25]. To activate such a platform in the platform layer, it is necessary to configure small platforms and connect the small platforms within the overall integrated platform [20,26]. Data integration for carbon neutrality can be achieved through the analysis of data connections between small platforms, such as the energy domain (Home, Building, PV/ESS, EV, etc.).

**Table 2.** Details of the Proposed System Architecture.

| Domain | Modules | Details | Related Service | Related Layer |
|---|---|---|---|---|
| Building | ✓ IoT Sensor/Gateway | The temperature/humidity/$CO_2$/fine dust sensors | ■ Distributed IoT-based Energy Data Integrated Management | Infra Layer |
| | ✓ Device Manager | Device management and operation | | |
| | ✓ Building Information Data Server | Building information data gathering and integrate management | ■ Energy Optimization Management Service Organization | |
| | ✓ Building Information Data Analyzer | Building energy analysis and management | ■ Demand Prediction | Big-Data Layer |
| | ✓ AI-based Building Demand Analyzer | Demand analysis and energy prediction | | Digital /Platform Layer |
| | ✓ Security and Authentication Management | User and EV identification and authorization management | ■ Security and Authentication Management | |
| PV/ESS | ✓ PV Manager | PV management and operation | ■ Supply Prediction | Infra Layer |
| | ✓ IBIE | Building integration main intelligent ESS | | |
| | ✓ PV (Inverter, PCS) | PV management and operation | ■ Peak Energy Management | |
| | ✓ IoT Sensor | Power generation (kWh), Charge (kWh), Discharge (kWh) data | ■ Distributed IoT-based Energy Data Integrated Management | |
| | ✓ PV/ESS Information Data Analyzer | PV/ESS energy analysis and management | ■ Supply Prediction | Big-Data Layer |
| | ✓ AI-based Building Supply Analyzer | Supply analysis and energy prediction management | | Digital /Platform Layer |
| | ✓ Energy Trading Management | User identification and authorization management | ■ Peak Energy Management ■ Smart Energy Trading | |

**Table 2.** *Cont.*

| Domain | Modules | | Details | Related Service | Related Layer |
|--------|---------|---|---------|-----------------|----------------|
| EV | ✓ | STE | Small temporary ESS for EVs | ■ Supply Prediction | Infra Layer |
| | ✓ | Charging Manager | EV charging integrates management and operation | ■ Smart Energy Trading | |
| | ✓ | Charging Station (CS) | Energy charging system for EVs | ■ Smart Energy Trading | |
| | ✓ | EV Information Data | EV information integrates management | ■ Data Connection | Big-Data Layer |
| | ✓ | User Information Data | User information integrates management | | |
| | ✓ | Energy Trading Management | Intelligent energy trading management system | ■ Smart Energy Trading<br>■ Security and Authentication Management | Digital /Platform Layer |

### 3.2. Overview of the System in the Building Domain

This section shows the establishment of a system for demonstration at the S Hotel in Seoul, Korea [24]. The status of the building consumer product and IoT packages are shown in Table 3. The total number of floors is 26 and consists of 383 guest rooms. The heating system is designed to operate 24 h a day. In regard to indoor air conditioning management, the customer operates the fan coil through the room temperature control device.

**Table 3.** Device and IoT Package Configuration in The Building.

| Num. | Item | Contents and Components |
|------|------|--------------------------|
| 1 | Testbed active optimal control system (IoT sensor) | - Room/Basement B1 Meeting Room Installation of IoT Sensor |
| 1-1 | | - Temperature–Humidity Sensor |
| 1-2 | | - Smart Integrated Gas (CO2) Sensor |
| 1-3 | Testbed active-type optimal control system (IoT sensor) components | - Smart Fine Dust Sensor |
| 1-4 | | - Occupancy Sensor |
| 1-5 | | - Smart Submeter (CT, Single Phase) |
| 2 | Testbed active-type optimal control system (control equipment) | - RCU Gateway |

The gateway and IEEE 802.11 wireless network protocol (Wi-Fi) repeater were assembled in a plastic enclosure and installed inside the ceiling inspection hole at the entrance to the room, and the temperature/humidity/CO2/fine dust sensors were installed on the shelf of the distribution box in the room by adding a protective cover. In addition, three smart submeters (current transformer (CT), single-phase power meter) were installed at the total power, electric heat, and light levels in the distribution box of the guest room. The smart occupancy sensor was installed in a location that can cover the entrance to the room and the interior.

Figure 5 shows the location and photo of the sensor in the actual building testbed. The sensor package was installed in the guest rooms on the 3rd, 15th, 25th, and 26th floors of the S Hotel and in the conference room on the 3rd basement floor. On the 3rd, 15th, 25th, and 26th floors, eight sensor packages were installed in each of the rooms, and one sensor package was installed in the waiting room and conference room on the 3rd basement floor.

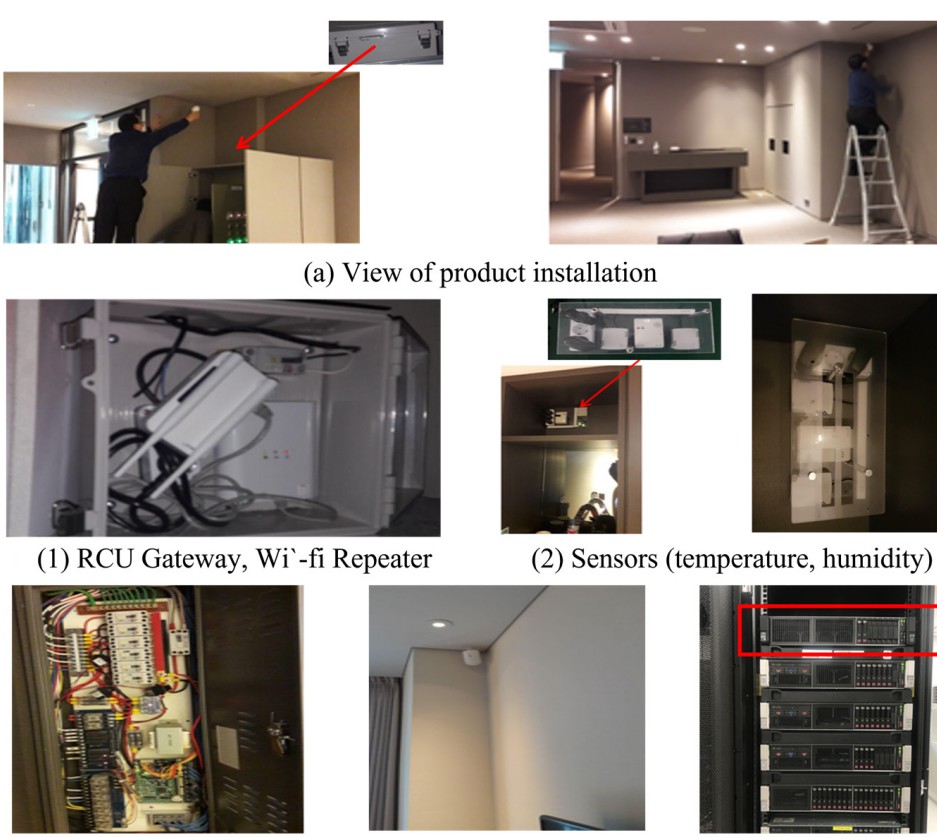

(a) View of product installation

(1) RCU Gateway, Wi`-fi Repeater   (2) Sensors (temperature, humidity)

(3) Smart submeter      (4) Occupancy sensor      (5) Main Server

(b) Configuration of IoT Packages

**Figure 5.** Sensor installation and device locations in the building.

### 3.3. Construction of PV and ESS Domain

Figure 6 shows the configuration diagram of the PV and ESS system constructed in the building [26]. The PV system is connected to the BEMS system constructed in the building, and data related to distributed power can be collected through IoT and AMI. The collected data are transmitted to the AI server through the gateway, and data-based optimization demand–supply forecasting analyses are performed. In addition, the data output from the built PV system can be monitored through PV simulation software, as shown in Figure 6. Through this simulation, it is possible to monitor the amount of solar power generated from the PV system and the amount of charging data. The PV and ESS platform plays a high-priority role in forecasting the energy supply. Further details are explained in the next section.

### 3.4. Construction of the EV/V2G Domain

In this paper, we propose an ESS-based EV as a critical energy source. In the EV, a portable battery is installed. Currently, it is simply applied to drive electric vehicles, but in a future society where electric vehicles are predominant, batteries built in electric vehicles will have a substantial role in energy sharing. Figure 7 shows a schematic diagram of the EV/V2G platform connected with BEMS and PV systems. Through IoT-based data collection, related data of EVs are collected and stored to the integration server.

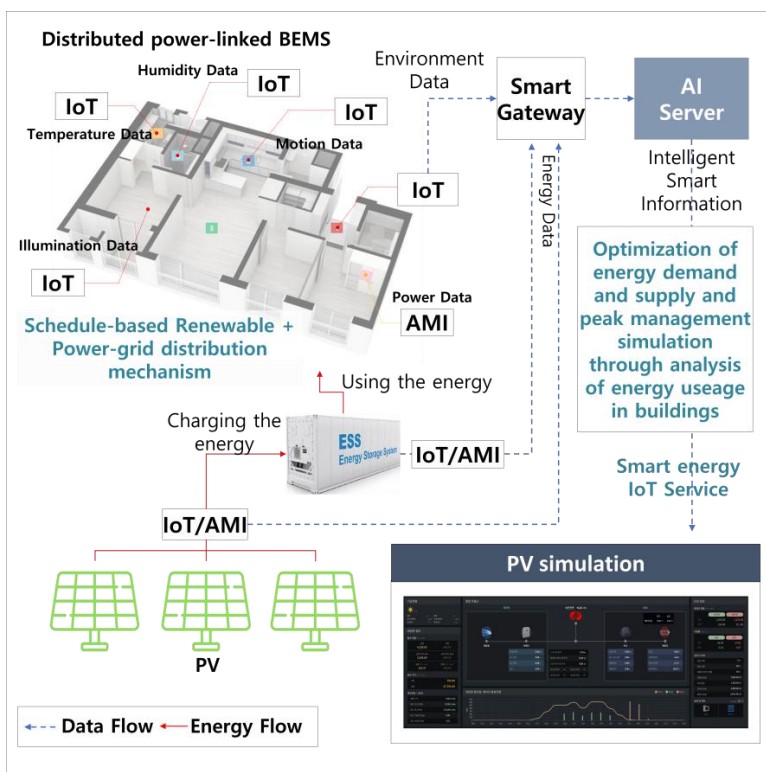

**Figure 6.** Construction of PV and ESS domain.

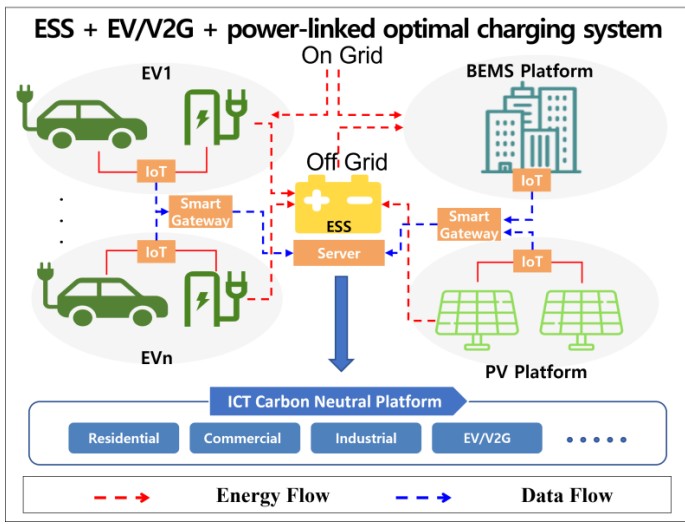

**Figure 7.** Construction of EV/V2G domain.

*3.5. Overall System Connection Diagram and Architecture*

Figure 8 shows the connectivity between the IoT package built in the building, the PV/ESS platform, and the EV/V2G platform. In the BEMS domain constructed in the building, temperature/humidity, user movement, fine dust, and power data are collected through IoT sensors and stored in the server through the gateway. Similarly, it is possible to collect related data based on IoT in the PV and EV platforms. The data related to PV/ESS consist of power generation data and charging data, and real-time data are utilized from 2019 to the present. The EV-related data include EV charging energy and discharging data and user schedule data.

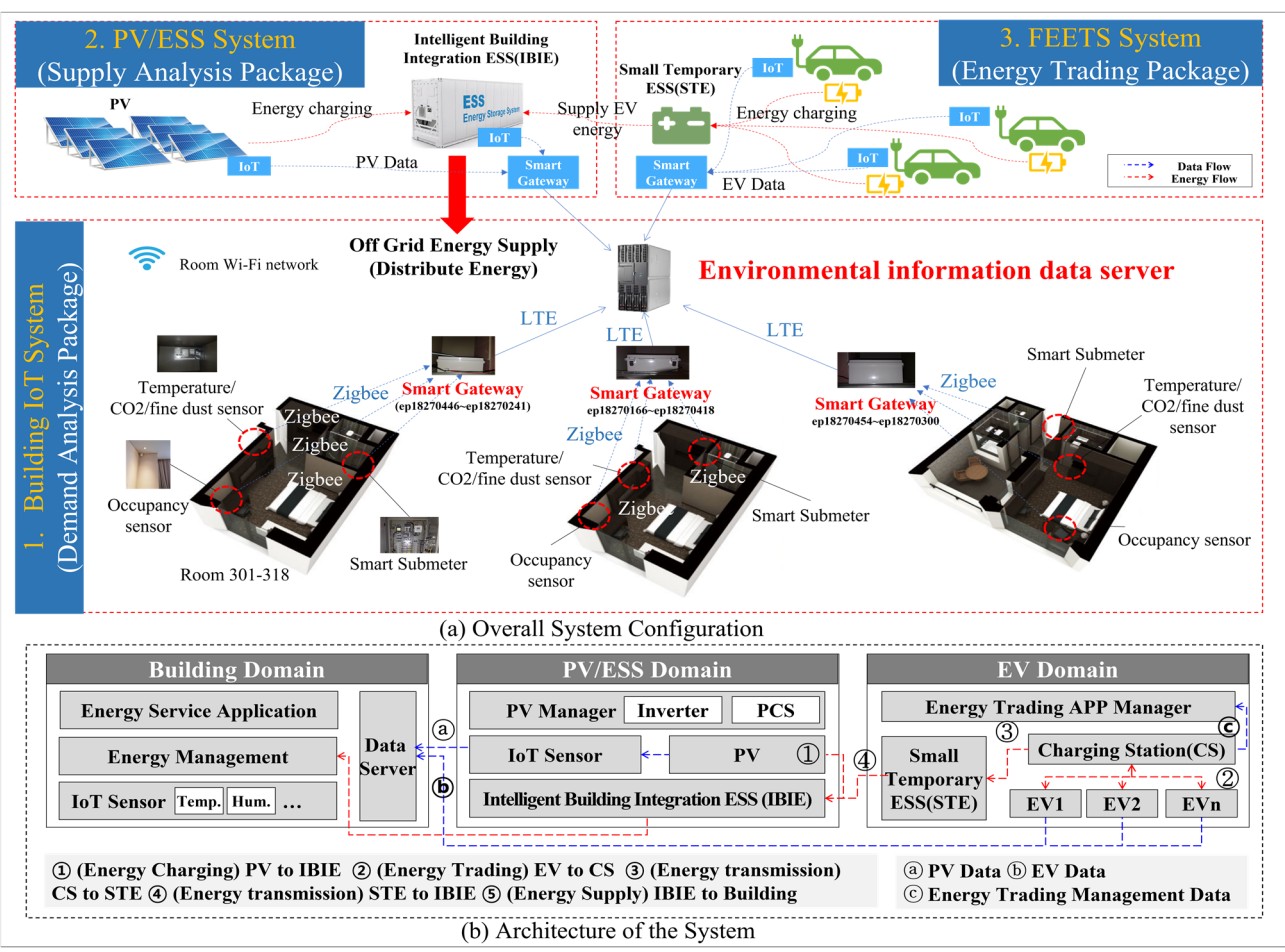

**Figure 8.** Connectivity between the IoT System built in the building, the PV/ESS platform, and the EV/V2G platform.

In this paper, EV-related data were implemented in the simulation, and the simulation was performed by building an actual futuristic EV platform testbed. An ESS is a system that stores energy and provides greater functionality than a conventional battery. First, an ESS can be divided into several types according to its utility. The predominant ESS in use is connected to distributed power in new and renewable energy. Notably, the battery used in an electric vehicle can be considered an ESS. To reduce the imbalance between energy demand and supply, the distributed grid should be accompanied by an ESS that can be used for charging in the event of excess power generation. Generally, it charges at off-peak times and discharges when demand is high. Therefore, the key elements of the proposed paper can be summarized as follows.

In Section 4, we show the mechanisms of building energy stabilization through supply and demand forecasting. Additionally, we designed a future EV/V2G system based on an energy sharing mechanism and scenario-based plan to prevent the building energy peak through the PV and EV systems in Section 5.

## 4. Methodology

### 4.1. Energy DR Management Methodology

Figure 9 shows the supply and demand forecasting methodology of the proposed system. Energy forecasting includes demand-side forecasting and supply-side forecasting. It is possible to predict buildings' energy demands based on the behavioral patterns of users inside buildings, energy usage patterns, and environmental information from environmental information sensors. Moreover, supply-side energy prediction consists of

the predicted energy generated from distributed power sources such as PV or renewable energy. When these energy demand and supply forecasts are efficiently made, it becomes possible to optimize the management of building energy stabilization and energy supply and demand efficiency.

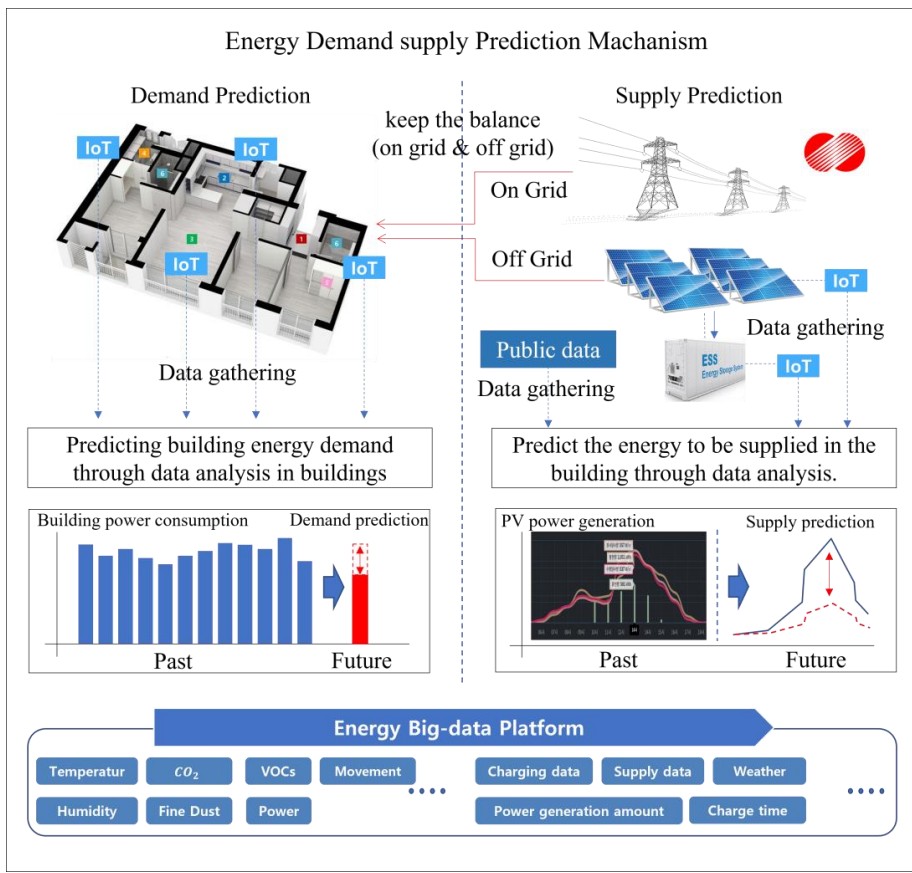

**Figure 9.** Building energy supply and demand mechanism.

### 4.1.1. Demand Prediction

Energy demand analysis is pivotal in analyzing future building energy consumption [27]. An energy supply and demand policy can be provided based on energy consumption forecast analysis. The energy demand forecast analysis can be analyzed utilizing smart IoT sensors constructed in the buildings. The environment can be analyzed through temperature/humidity, motion, fine dust, and power sensors. Then, energy demand can be predicted through user and environmental pattern analysis [28].

### 4.1.2. Supply Prediction

Supply-side predictive analysis can predict the amount of energy supplied for energy obtained from renewable energy [29]. This is because an optimized energy policy can be established only by analysis of the amount of energy demanded and the amount of supply. For example, if it is predicted that the building's peak energy will occur through energy demand analysis, energy must be supplied from PV and distributed power sources. However, if it is difficult to supply even when utilizing distributed power, other energy sources will have to be found. Therefore, in this paper, we propose an energy supply and transaction mechanism that can find an energy supply strategy through building energy supply and demand forecasting. In addition, the PV/ESS and EV/V2G platforms are connected as a core platform for energy supply. While EVs are currently valuable as a means of transportation, their future value lies in not only providing a means of transportation but also in provisioning a mobile ESS. Therefore, the EV platform is fundamentally important for a mobile ESS.

*4.2. Three Steps to Avoid Building Peaks (BEMS, PV/ESS, and EV Demand–Supply Management)*

Figure 10 shows a three-step energy supply strategy when a building energy peak occurs. Step 1 can be divided into Case 1 and Case 2. Case 1 depicts that the appropriate energy demand is predicted, and Case 2 shows where an energy peak in a building occurs. In Case 2, energy charges are generated according to peak energy, so an energy supply strategy must be established accordingly. As a supply strategy, in Step 2, PV/ESS platform-based energy supply forecasting is enacted.

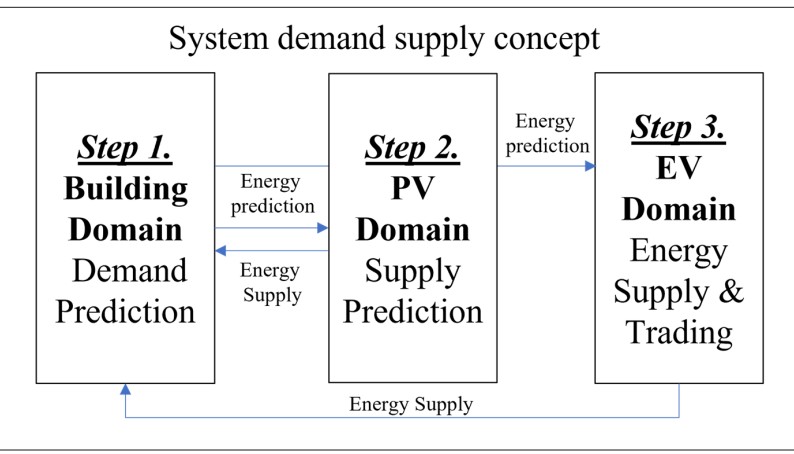

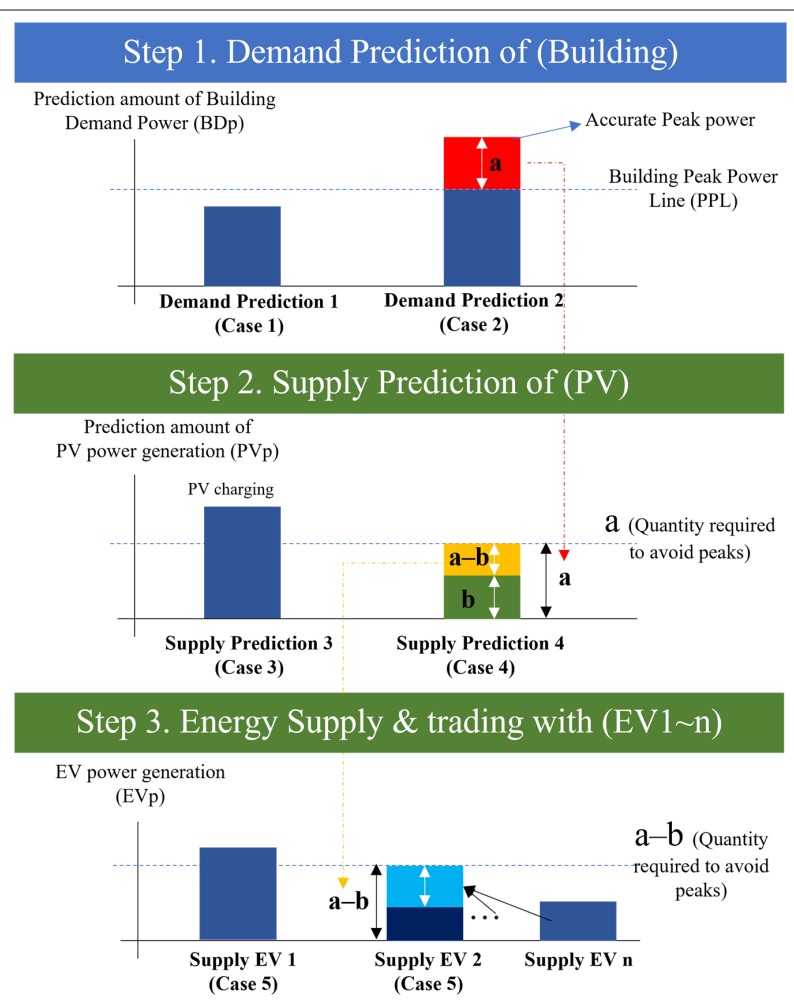

**Figure 10.** Energy supply strategy of the building's energy peak.

Case 3 depicts a situation that predicts that a large amount of energy is expected to be charged to the ESS by producing adequate energy from the PV system. Case 4 highlights circumstances in which the energy supply and demand is predicted to be difficult due to various environments. In Case 3, since the supply is larger than the building's energy peak demand (a), it is sufficient to supply as much energy as the peak power demand. In Case 4, the supplied energy is less than the peak energy demand, so the energy of a–b must be supplied from another source. In this case, the supply is made through the EV/V2G platform-based energy transaction in Step 3. This structure collects and supplies scattered EV/ESS energy so that it provides as much energy as needed by a–b, which cannot be supplied from the PV system.

### 4.3. Platform Connection-Based Energy Management Algorithm

#### 4.3.1. Step 1. Demand Prediction of the Building

Figure 11 shows a three-step energy supply strategy when a building energy peak occurs. In the demand forecasting step (Step 1), the demand analysis of the building is performed, and BDp represents the predicted demand amount in the building. In addition, PPL represents the building's peak limit line. Importantly, if BDp is greater than PPL, it means that a building energy peak will occur. If an energy peak occurs through demand analysis, a corresponding supply strategy is required, so a PV supply forecast is performed through the supply forecasting step (Step 2).

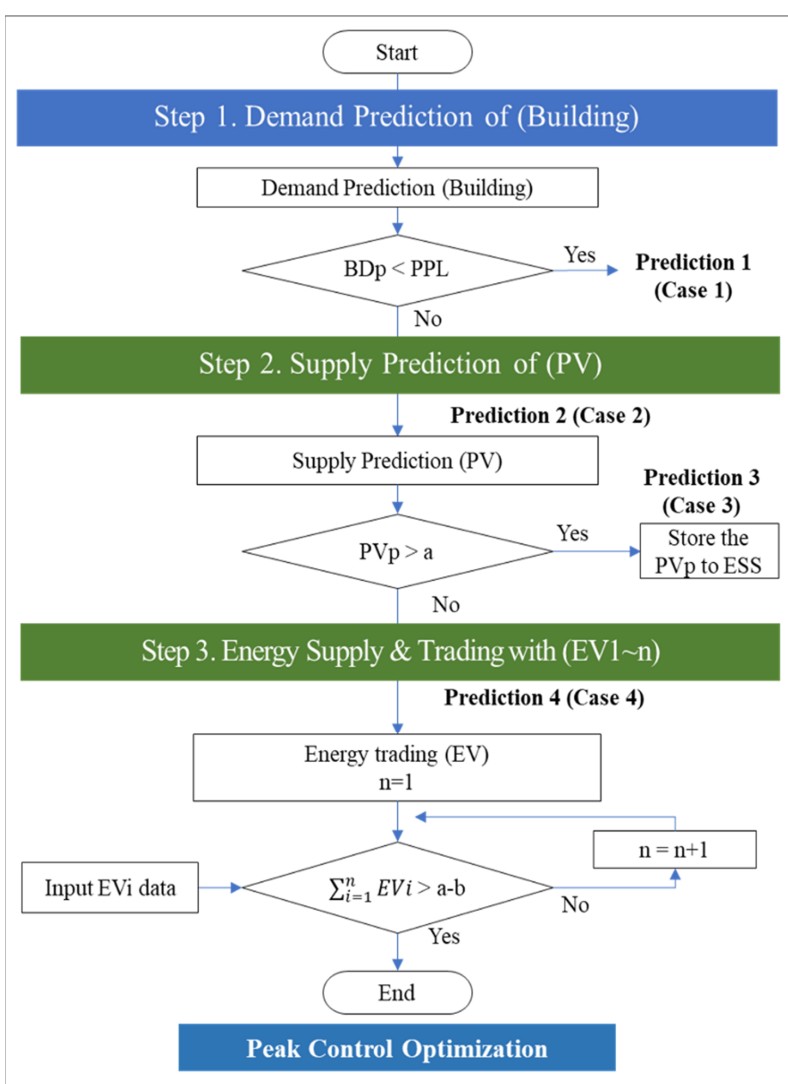

**Figure 11.** Three-step energy supply strategy algorithm.

### 4.3.2. Step 2. Supply Prediction of the PV and ESS

If the prediction in Step 1 indicates that a power peak in the building will occur, the energy supply strategy should be established through the PV platform [30]. Therefore, in Step 2, a strategy is established to supply the PV power to the building's ESS. If PVp is greater than a, it means that the peak can be prevented by supplying energy from the amount provided by PV power generation. As shown in Figure 11, if PVp is less than a, another source must be found that can supply the power for a–b. In this paper, we set the source of the next step as the EV platform.

### 4.3.3. Step 3. Energy Supply and Trading with EV

As in Step 2 of Figure 10, the power of a–b is supplied through the energy trade with the EV. The surplus power of EVs is collected and stored in the building's main ESS. In Step 3 of Figure 11, EVi represents the amount of electricity traded by the electric vehicle. When $i = 1$, EV1 is the first energy trading EV. The sum of all power transactions can be expressed by the following formula.

$$\sum_{i=1}^{n} EVi \tag{1}$$

When the sum of all electricity transactions is greater than the amount of the electricity of a–b, the transaction is completed.

## 5. Proposed System Scenarios

Figure 12a shows a schematic diagram of the entire scenario. It is divided into three domains: building, PV + ESS, and EV(FEETS) domains. In the scenario of the proposed system, the EV domain scenario considers a futuristic building parking system. The futuristic parking energy supply system is subsequently described in detail.

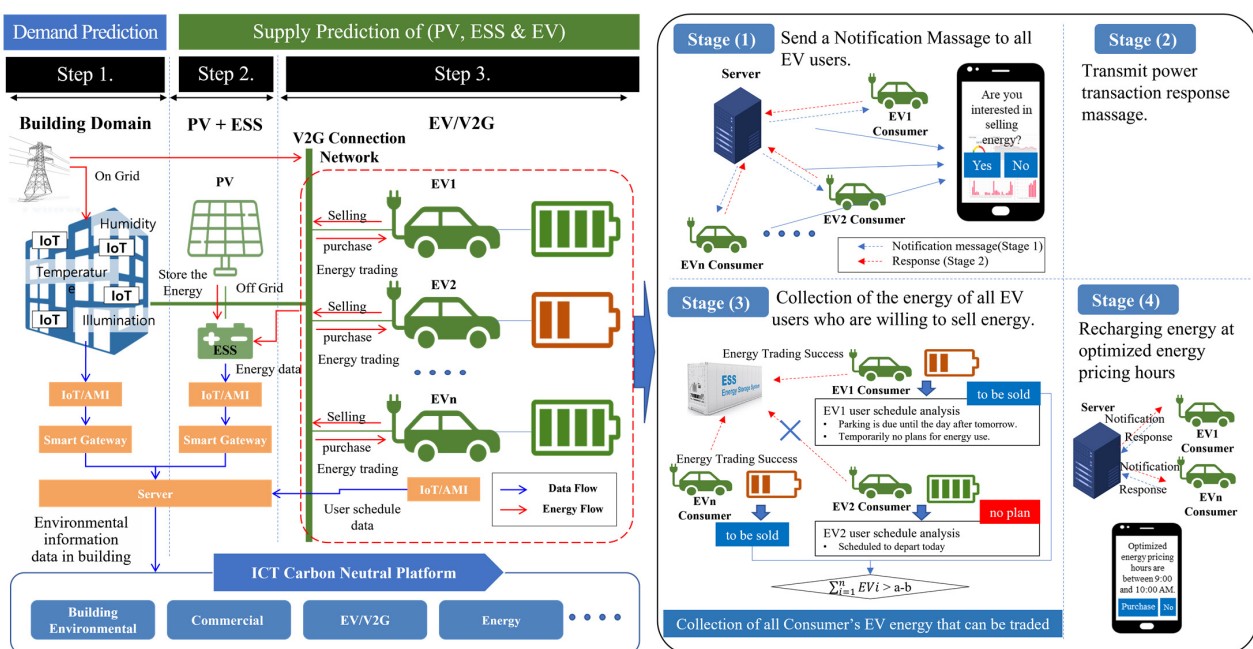

(a) Scenario configuration overview of the proposed system  (b) System application flow in FEETS for energy trading(in Step 3)

**Figure 12.** Building with PV and ESS connections and a FEETS overall Scenario.

### 5.1. Step 1. Building Domain

In the building domain, smart IoT sensors deployed in buildings collect environmental information and transmit it to a smart gateway [24]. The gateway sends these data to the server. As shown in Figure 12a, each data point is stored as part of an energy big-data platform. The server calculates the energy demand forecast and receives energy through

PV and EV sources according to the proposed energy DR management methodology, as previously described in Section 4.

### 5.2. Step 2. PV + ESS Domain

In the PV + ESS domain, similar to the building domain, IoT sensors collect energy data such as the power generation and ESS charging amount [26,30]. As shown in Figure 12a, the data are then stored to the energy big-data platform of the server through the gateway. The PV system produces the energy based on the predicted building energy demand calculated in the building domain. The EV platform contains numerous irregular elements. Since PV generation is affected by weather, it is important to analyze environmental factors.

### 5.3. Step 3. EV Domain (FEETS_Energy Transaction)

In this paper, we propose a FEETS, a futuristic parking energy supply and demand system [11]. This leading-edge parking energy supply system assumes that in the future, almost all cars will be converted to EVs, and magnetic-based wireless EV charging systems will be installed in all building parking lots. This means that all EVs can be charged and traded at any time during their parking sojourn. Figure 12b shows the system application flow for energy trading between the main BEMS server and EVs. When starting an energy transaction between the BEMS and EV, the server sends a query to each EV user in the parking lot regarding the energy trading (Stage 1). As shown in Figure 12b, users receive the sent query through a smartphone application. The EV user clicks Yes to make the transaction and the server collect all EV energy that can be traded in energy (Stage 2, 3). Subsequently the server sends an energy purchase query message considering the optimal energy purchase price to users who have traded energy (Stage 4). In conclusion, BEMS operators obtain economic benefits by collecting insufficient energy from EVs, and EV users can purchase energy at a low price, resulting in a win-win situation for each other.

### 5.4. System Connection Process Based on the FEETS Overall Scenario

Figure 13 shows the proposed system data and energy connection energy flow [26].

The top of Figure 13 shows the overall system configuration, and the middle and bottom show the data linkage and energy flow status. As shown on the left of Figure 13, divided into three parts are: (1) Data Connection Flow, (2) User security and authentication, and (3) Application Data Connection Flow. In the Data Connection Flow part, it shows a data connection diagram for Demand Prediction and Supply Prediction presented in Section 4.1. The IoT sensor collects data and transmits it to the server, which shows all data flows of the energy demand side and supply side. In the User security and authentication part, it shows the overall flow of user authentication and security in energy transactions. The identification and authorization management of the user and EV network is a very important factor in energy trading for user satisfaction and safety. Application Data Connection Flow part shows the application-based system flow presented in four stages in the FEETS overall Scenario of Figure 12.

### 5.5. Business Model of FEETS

Figure 14 shows the overall system BM configuration and detailed scenario of the proposed system. The side view shows the building, PV and ESS, and the futuristic EV parking charging and trading system at the first and second basement floors. The floor plan view shows the detailed structure of the futuristic EV parking, charging, and trading system. As explained earlier, this proposed system is a futuristic system, and it is assumed that all cars will be converted to EVs and all sectors of parking lots will be built with wireless charging systems [31]. When a peer-to-peer (P2P)-based energy transaction is performed between the energy trading system and the EV, the energy in the EV is stored to the small temporary ESS (STE). The energy of the STE is stored to the main ESS in the BEMS. This energy will be used to fill the peak power in the building. In addition, the data (the energy charge amount, discharge amount, and user schedule information for each

user EV) are collected through the IoT sensor and transmitted to the gateway. The gateway collects all data and sends them to the main server. The flows of data and energy are shown in Figure 14.

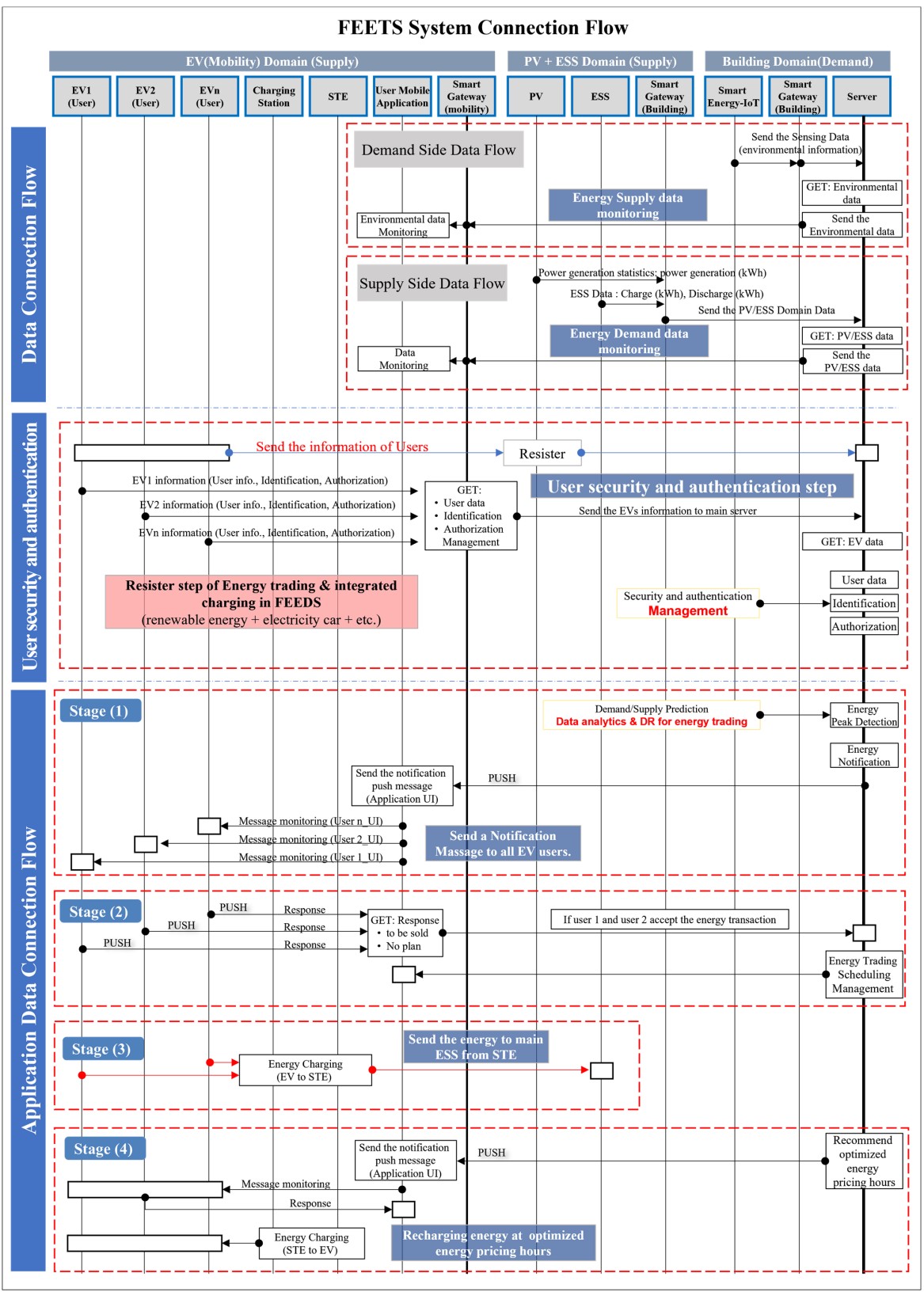

**Figure 13.** FEETS System configuration and energy, data connection flow.

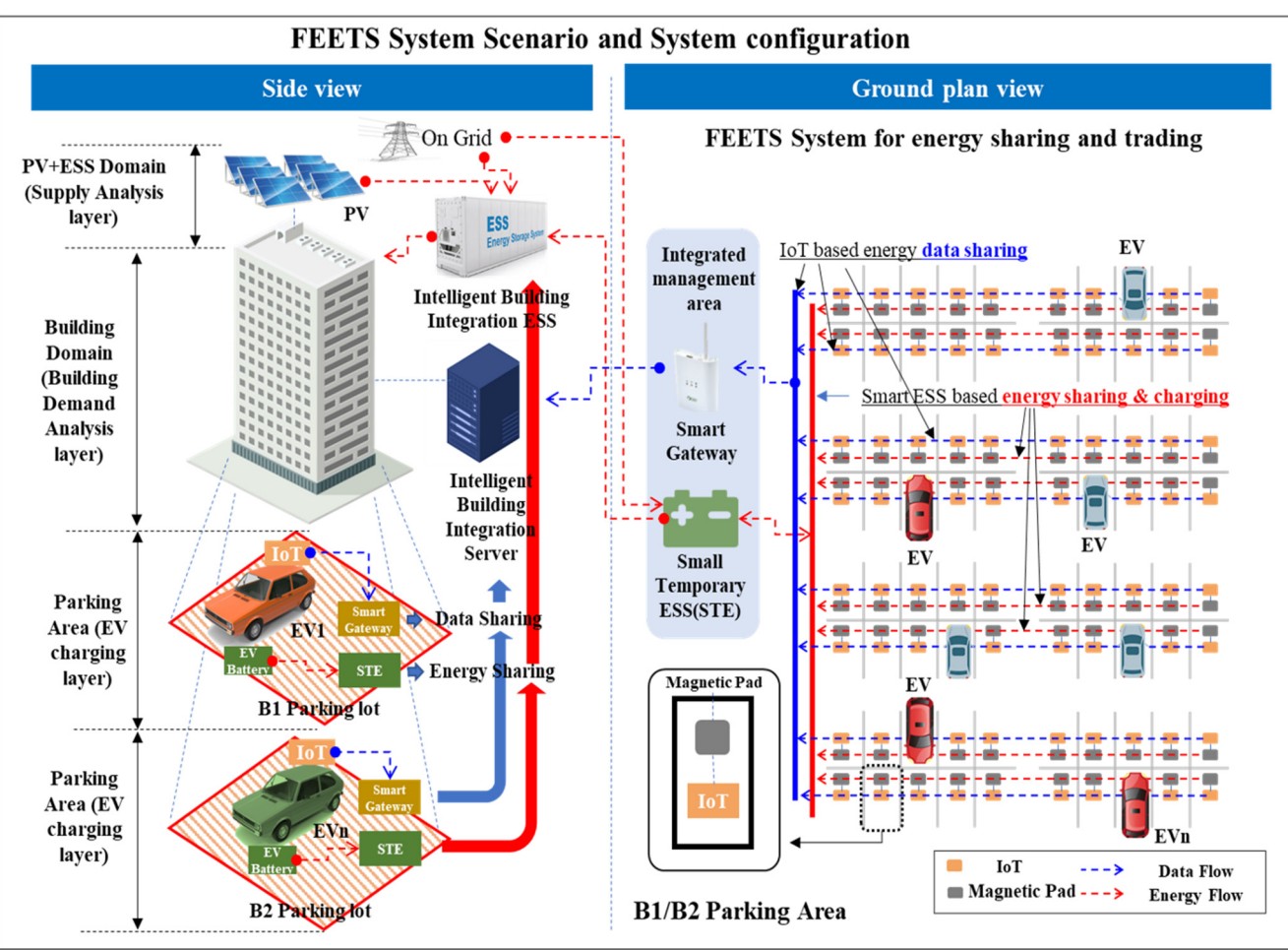

**Figure 14.** FEETS System configuration and energy, data flow scenario.

## 6. Business Analysis Simulation Based on a Scenario

In Section 6, we introduce the Korea's energy policy and contract power and analyze the electricity bill for one month through scenario-based simulations of an existing building and the proposed system.

### 6.1. Simulation Value Settings

Table 4 shows the value settings for the scenario simulation. This information consists of virtual value settings to provide a logical explanation of the proposed technology. In the building domain noted in Table 4, the contract power is straightforward when the usage is within the contract power amount, but if more power is used than that stipulated in the contract, additional charges are incurred, which is demonstrated in the following Table 5 [32]. Contract power policies have been enacted in Korea to allow building owners to set up a specific contract for power, thus inducing energy savings by granting an additional amount when excess power is generated [33].

Therefore, the proposed system aims to reduce energy based on the PV/ESS and EV/V2G connection to prevent such excess energy. This simulation compares and analyzes the economic feasibility of electricity bills for one month between the existing building without the proposed system and the building with the proposed system.

**Table 4.** Value Settings for The Proposed System Scenario Simulation.

| Num. | Domain | Scenario Parameter | | Value | |
|------|--------|--------------------|---|-------|---|
| 1 | Building | - Past building electricity per month (BE)<br>- Contract power (CP)<br>- CP per month | | - 181,175 kWh<br>- 200 kWh<br>- 144,000 kWh | |
| 2 | PV/ESS | - PV capacity per unit<br>- Num. of PV<br>- ESS capacity | | - 0.5 kWh<br>- 278<br>- 1 MWh | |
| 3 | EV | - EV battery capacity per unit | | - 100 kWh | |
| 4 | Environmental information | - Season | | - Winter | |
| 5 | Power type | - Power type | | - Low-pressure power | |

**Table 5.** Criteria For Applying a Contract Power Excess Charge [32].

| Domain | Scenario Parameter |
|--------|--------------------|
| 451 kWh/month<br>~<br>kWh/month | - Excess power consumption (EPC) × base rate unit price(B) × 150%<br>- Excess power consumption (EPC) × base rate unit price(B) × 200%<br>- Excess power consumption (EPC) × base rate unit price(B) × 250% |
| ~720 kWh/month excess | - Excess power consumption (EPC) × base rate unit price(B) × 300% |

### 6.2. BM Analysis Simulation Procedure Based on a Scenario

#### 6.2.1. Existing Building Electricity Bill Simulation

Figure 15 shows an example of a demand simulation to illustrate power peak occurrences for contract power in an existing building. The total electricity charge (TEC) is calculated as the basic charge (BC), the electricity charge (EC), and the excess charge (EXC), and the calculation method is as follows:

$$\text{CP per month} = \text{CP} \times 24 \text{ [time]} \times 30 \text{ [day]} \tag{2}$$

$$\text{BC} = \text{CP} \times \text{A} \tag{3}$$

$$\text{EC} = \text{BE} \times \text{B} \tag{4}$$

$$\text{EXC} = \text{EPC} \times \text{B} \times \text{increase rate (\%)} \tag{5}$$

$$\text{TEC} = \text{BC} + \text{EC} + \text{EXC} \tag{6}$$

The following shows the electricity rates calculated based on an example of the scenario simulation. When calculating with reference to the example data shown in Table 6, BC is calculated as KRW 1,232,000 and EC as KRW 18,045,030 by Equations (3) and (4) as follows:

$$\text{BC} = \text{KRW } 1,232,000 = 200 \text{ kWh} \times 6160 \tag{7}$$

$$\text{EC} = \text{KRW } 18,045,030 = 181,175 \text{ kWh} \times 99.6 \tag{8}$$

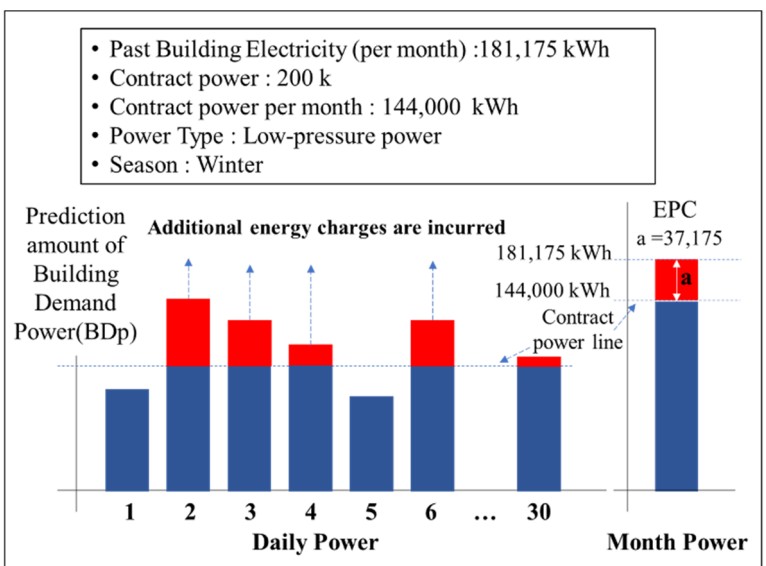

**Figure 15.** Existing building electricity amount by day and month.

**Table 6.** Standard Electricity Charge for General Use (Won/kwh) [34].

| Power Type | | Base Rate Bill (KRW) (A) | Electricity Bill (Month) (B) | | |
|---|---|---|---|---|---|
| | | | Summer (6~8) | Spring/Fall (3~5, 9~10) | Winter (11~12) |
| Low-pressure power | | 6160 | 113.0 | 72.5 | 99.6 |
| High-pressure power A | I | 7170 | 123.2 | 79.2 | 110.9 |
| | II | 8230 | 119.2 | 74.9 | 105.6 |
| High-pressure power B | I | 7170 | 121.1 | 78.1 | 107.9 |
| | II | 8230 | 115.8 | 72.8 | 102.6 |

If the contract power (CP) is 200 kw, the contract power per month is calculated as 144,000 kWh by Equation (2). Excess power consumption (EPC) refers to the difference between the amount of BE and the CP per month. Therefore, the EPC is calculated as 181,175 kWh − 144,000 kWh = 37,175 kWh. The EPC is 37,175 kWh; when referring to Table 5, the increase rate (%) is 300%. Therefore, the EXC is calculated as follows.

$$\text{KRW } 11{,}107{,}890 = 37{,}175 \text{ kWh} \times 99.6 \times 300\% \tag{9}$$

This finding shows that a large amount of the charge is due to contract power. Therefore, calculating the total electricity charge for one month shows that it is 30,384,920 won (KRW 1,232,000 + KRW 18,045,030 + KRW 11,107,890). This is the worst-case scenario if the contract power policy is applied.

6.2.2. Proposed Building Application with a System Electricity Bill Simulation

Figure 16 is a scenario-based simulation graph of each situation. Table 7 shows six situations that consider the building demand, PV supply, and weather influences. Table 7 indicates that Situations three to six require the management of peaks. In addition, Situations four and six require that more energy needs to be provided by another energy source. In this paper, the electricity bill was compared and analyzed for one month to determine the economic feasibility of both the existing building and the building utilizing the proposed system. To analyze the electricity bill, we calculated the worst and best cases in the proposed building system. The worst case reflects an entire month of cloudy days, and in the best case, the entire month consists of sunny days.

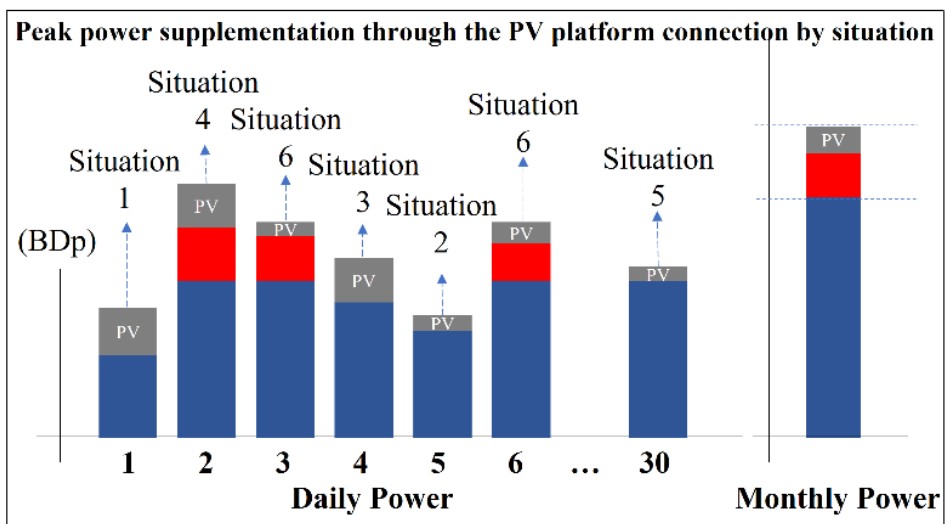

**Figure 16.** Proposed building situations by day and month.

**Table 7.** Situational Characteristics of The Proposed System Scenario Simulation.

| Class | Building Demand Rate | PV Supply Rate | Weather | Situation Details |
|---|---|---|---|---|
| Situation 1 | Low | High (100%) | Sunny | - No peak affect area |
| Situation 2 | Low | Low (10%) | Cloud | - No peak affect area |
| Situation 3 | High | High (100%) | Sunny | - Peak affect area<br>- Energy supply through PV<br>- Does not require another energy source |
| Situation 4 | High | High (100%) | Sunny | - Peak affect area<br>- Energy supply through PV<br>- Requires another energy source |
| Situation 5 | High | Low (10%) | Cloud | - Peak affect area<br>- Energy supply through PV<br>- Does not require another energy source |
| Situation 6 | High | Low (10%) | Cloud | - Peak affect area<br>- Energy supply through PV<br>- Requires another energy source |

In the best case, the energy supplied by the PV, zone b in Figure 17, increases. Moreover, in the worst case, the energy of zone b decreases. Therefore, the total electric vehicle energy trading price (EV-ETP) will be determined according to the weather. In Equation (10), the addition of the EV energy transaction price is shown in terms of the total electricity charge (TEC) of the proposed system.

$$TEC_p = BC + EC + (EV - ETP) \qquad (10)$$

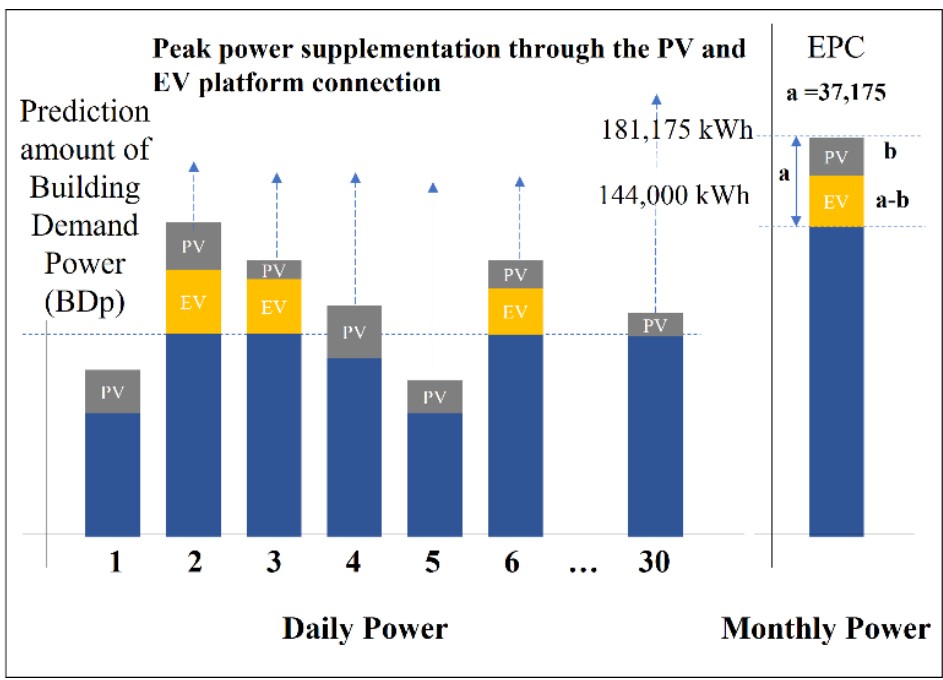

**Figure 17.** Peak power supplementation through the PV and EV.

In this paper, the TEC values were compared by dividing the worst case and the best case. As discussed, the best case depicts an entirely sunny month. On a sunny day (best day), it is assumed that the PV supply is 100%; on a cloudy day (worst day), it is assumed that the PV supply is 10%, and the PV generation for one month is as follows.

- One day (10~14:00) production per PV panel (0.5 k): 3 k.

Considering the roof area of the building, 278 panels can be installed, and the PVs on a clear day and PVc on a cloudy day are as follows:

- Best-case PV generation amount per day (PVs): 834 kWh;
- Worst-case PV generation amount per day (PVc): 83.4 kWh (Set to 10% of PVs).

The following shows the range of electricity consumption between the PVs and PVc per month.

$$PV_s = 834 \text{ [kWh]} \times 30 \text{ [day]} = 25{,}020 \text{ kWh} \tag{11}$$

$$PV_c = 83.4 \text{ [kWh]} \times 30 \text{ [day]} = 2502 \text{ kWh} \tag{12}$$

Table 8 shows the situational electric vehicle charging power bill and the comparison of EV energy transaction prices in each situation (best case and worst case) is as follows.

**Table 8.** Situational Electric Vehicle Charging Power Bill (Won/Kw) [35].

| Class | Time | Summer (6~8) | Spring/Fall (3~5, 9~10) | Winter (11~12) |
|---|---|---|---|---|
| | | **Electricity Bill (KRW)** | | |
| Low-pressure power | 23:00~09:00 | 64.9 | 66.0 | 88.0 |
| | 09:00~10:00 12:00~13:00 17:00~23:00 | 152.6 | 77.8 | 135.5 |
| | 10:00~12:00 13:00~17:00 | 239.8 | 82.7 | 198.1 |

In Figure 18, the best case, the amount of electricity to be supplied from the EV per month is 37,125 kWh − 25,020 kWh = 12,105 kWh, and when converted into price (KRW), it is 2,398,000 won.

$$EV - ETPs = 12,105 \text{ kWh} \times 198.1 = (KRW) \, 2,398,000 \tag{13}$$

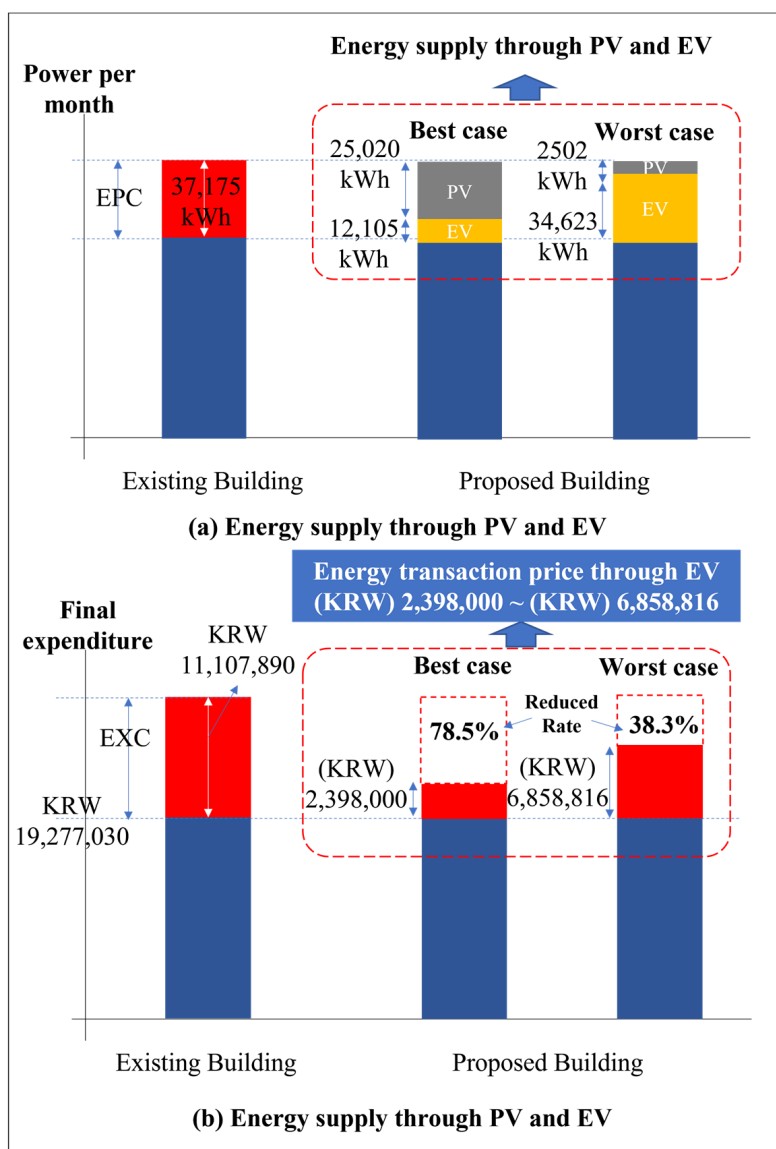

**Figure 18.** Existing building electricity bill by day and month.

In addition, in the worst case, the amount of power to be supplied from the EV is 37,125 kWh − 2502 kWh = 34,623 kWh, and when converted into price (KRW), it is 6,858,816 won.

$$EV - ETPc = 34,623 \text{ kWh} \times 198.1 = (KRW) \, 6,858,816 \tag{14}$$

Therefore, the transaction price from EVs in the best-case and the worst-case ranges from (KRW) 2,398,000 to (KRW) 6,858,816, respectively. Comparing it with the EXC of KRW 11,107,890 calculated in the existing building, a large amount of cost is reduced. EV users will have economic benefits since energy can be sold at a high price and energy can be bought at a low price.

*6.3. Business Model of FEETS and Future Platform Strategy*

6.3.1. The Used Energy Trading Platform (UETP)

The proposed system is based on a future carbon-neutral society. Currently, the used market is active in Korea. Based on these consumers' sales psychology, the energy trading business model can be formed under the name of a used energy trading platform (UETP) to share the surplus energy of electric vehicles for each consumer's appropriate profits in the future. Through this UETP platform, consumer A can sell surplus energy to other consumers who need energy, and then consumer A can recharge energy at optimized energy pricing hours. As shown in Figure 12, this platform proceeds based on user applications. If this energy trading platform is activated, it is expected that it will become an innovative service as a platform for buying and selling surplus energy among the consumers in the future. At this point, it is necessary to expand the supply of electric vehicles and intelligent electric charging stations at the national level.

6.3.2. Importance of Small Platform Connection

In the future, the importance of systems based on small platform connection will increase. This paper also explains the connectivity between buildings, PV, and EV platforms. It is important to expand the energy business model and provide services to achieve carbon neutrality through connection between small platform and small platform. We expect that such platform connection and data sharing can be achieved through city- or national-based platforms such as the carbon-neutral digital innovation platform presented at the beginning.

**7. Conclusions**

In this paper, we introduced a carbon-neutral digital innovation platform that aims to achieve carbon neutrality in 2030 and proposed a supply mechanism for energy demand forecasting in BEMS by using this platform. Given the importance of understanding the rationale behind energy supply and demand predictions, this prediction method leveraged possibilities to establish energy stabilization policies in various domains, such as buildings, factories, and homes. We also proposed PV/ESS and EV/V2G as alternatives for energy stabilization and examined how energy peak and stabilization measures can be established through the connection of the platforms. Furthermore, a futuristic business model and smart PV and EV charging and trading platform were introduced. As such, the connection of each platform is important for carbon neutrality. In the future, various business models, services, and studies will be needed for 2030 carbon neutrality.

**Author Contributions:** Conceptualization, S.P. (Sehyun Park), S.P. (Sangmin Park), B.K. and M.-i.C.; data curation, S.P. (Sangmin Park), B.K. and M.-i.C.; formal analysis, S.P. (Sangmin Park); methodology, S.P. (Sehyun Park), S.P. (Sangmin Park), S.P. (SeolAh Park), S.-P.Y., K.L., B.K., M.-i.C. and H.J.; project administration, S.P. (Sehyun Park); supervision, S.P. (Sehyun Park); validation, S.P. (Sangmin Park) and S.P. (SeolAh Park); writing—original draft, S.P. (Sangmin Park) and S.P. (SeolAh Park); writing—review and editing, S.P. (Sangmin Park) and S.P. (SeolAh Park). All authors have read and agreed to the published version of the manuscript.

**Funding:** This work was supported by the Human Resources Development (No. 20214000000280) of the Korea Institute of Energy Technology Evaluation and Planning (KETEP) grant funded by the Korea government Ministry of Trade, Industry and Energy, and this research was supported by the Chung-Ang University Research Scholarship Grants in 2019.

**Data Availability Statement:** Data are contained within the article.

**Conflicts of Interest:** The authors declare no conflict of interest.

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
