# Peer review of "Design and Implementation of a Futuristic EV Energy Trading System (FEETS) Connected with Buildings, PV, and ESS for a Carbon-Neutral Society"

_buildings, doi:10.3390/buildings13030829_

Round 1

Reviewer 1 Report

Brief summary

The energy sector, as structured in the present, is dependent on traditional energy resources such as oil and coal. These resources are polluting by issuing large amounts of greenhouse gases into the atmosphere. On the other hand, the price of such resources is very sensitive to geopolitical stresses in the extraction areas. These are the reasons why it is desired to reach carbon neutrality and find alternative solutions for traditional energy resources. Carbon neutrality is not an easy goal to achieve. The authors analyze the energy consumed and propose an energy sharing scenario for energy stabilization because this sector is one with a very high energy consumption, having consumption peaks in certain time intervals also prevent excess prices. In the article A Distributed IoT-based Energy-Management Building Framework with PV and ESS Connections and a Futuristic EV Energy Trading System (FEETS) for Carbon Neutral Society the authors use current energy and telecommunications technology like Internet of Things (IoT) to introduce a distributed-IoT sensor constructed in the building and explains the mechanism of demand forecasting in the building and supply forecasting in the photovoltaic (PV) system.

Strengths and weaknesses

For supplying the energy needed for internal consumption in a building (electronic equipment), the photovoltaic system is not sufficient, being dependent on the weather (duration of sunshine). For an energy supplement, the authors propose an alternative energy source: electric vehicles, an efficient energy supply among the building energy management system (BEMS). By using mixed integer linear programming, two-way energy trading capacity of electric vehicles can be managed and this is strength of the article. To be convincing, the authors design the Carbon Neutrality Platform for a hotel, a case study in which they collect data and look for solutions to cover the peaks energy consumption in buildings and this is strength of the article. I have not identified any weaknesses. I believe that the article covers the entire problem in the field of energy consumption in buildings and can be published.

Author Response

We thank you for your comment and appreciate informing us for our paper.

We are very grateful for the reviewer's evaluation of the paper. We have revised the paper a little more, and we will do our best to conduct better research in carbon neutrality.

We have revised the introduction (page 1~4), related work (page 4~5) chapters and added the algorithm for the whole method (page 17~18). please check the revised version and thank you very much for the reviewer's comments and evaluation. If there is anything that needs to be corrected, please inform us again and we will apply it right away.

Reviewer 2 Report

The title is too long and complex. Consider rewriting the title - avoid using all keywords in the tile.

Line 37: It is already past 2020. So, the authors should consider presenting the actual figures for 2020 and not the forecast. 

Paragraph 1 is too long. Please consider breaking it down into at least 2 to 3 paragraphs.

The introduction and literature review are not very rich. More relevant literature needs to be studied and reported. Moreover, the bride literature review is just a single paragraph composed of statements stating what other works have done. There is a need for a more critical review.

The section on system architecture seems like a stand-alone section. It started with a Figure showing the architecture of the proposed system. It is unclear if the authors first proposed this architecture or adopted it from earlier works. Moreover, the section was not introduced properly. It is too short to be treated as an entire paper section. 

Would combining sections 2 and 3 into a single section with subsections be better?

In Figure 3, the legend indicates both energy and data flow. However, the configuration diagram does not have any data flow within it. Is this the case, or is it an error? Kindly check and update.

Figure 4 has different line styles. But there is no legend to interpret the different line styles. Hence, it is difficult to comprehend the diagram.

Some critical elements of the paper are summarised in lines 205 to 212. These should be part of the introduction, except if there is a reasonable justification for the current position.

Author Response

Point 1: The title is too long and complex. Consider rewriting the title - avoid using all keywords in the tile.

Response 1: We thank you for your comment and appreciate informing us.
We have modified the title based on the reviewer's comments. We think that the title is very difficult, too long, so we correct the title. Please refer atteched note file. Thank you very much for the reviewer's comments for our paper.

Point 2: Line 37: It is already past 2020. So, the authors should consider presenting the actual figures for 2020 and not the forecast. 

Response 2: We thank you for your comment and appreciate informing us.
This sentence shows the carbon emissions of the building sector in the past year 2020. So, please note that these figures are not predictive of the future in 2020. This numbers are from the actual past figures in 2020. 
Thank you very much for the reviewer's comments. If there is anything that needs to be corrected, please inform us again and we will apply it right away.

Point 3: Paragraph 1 is too long. Please consider breaking it down into at least 2 to 3 paragraphs.

Response 3: We thank you for your comment and appreciate informing us. We have corrected the Paragraph 1 down into at least 2 to 3 paragraphs. 
Thank you very much for the reviewer's comments. If there is anything that needs to be corrected, please inform us again and we will apply it right away.

Point 4: The introduction and literature review are not very rich. More relevant literature needs to be studied and reported. Moreover, the bride literature review is just a single paragraph composed of statements stating what other works have done. There is a need for a more critical review.

Response 4: We thank you for your comment and appreciate informing us. We have corrected the literature review. Please refer atteched note file. And we have added the “Related Work” chapter in the paper. 
Thank you very much for the reviewer's comments. If there is anything that needs to be corrected, please inform us again and we will apply it right away.

Point 5: The section on system architecture seems like a stand-alone section. It started with a Figure showing the architecture of the proposed system. It is unclear if the authors first proposed this architecture or adopted it from earlier works. Moreover, the section was not introduced properly. It is too short to be treated as an entire paper section. 

Response 5: We thank you for your comment and appreciate informing us. We have corrected the section of the system architecture.
This carbon-neutral platform is our main research subject that is being researched in our lab. So, we add the architecture in the “3.1. System Architecture”. And in the chapter “3.1. System Architecture” we have presented the chapter “3.1.1. Carbon-Neutral Digital Innovation Platform (CNDIP)” and chapter “3.1.2. Proposed System Architecture (Connection of CNDIP)”.
We have subdivided the contents of Chapter 3 by merging and separating the chapters.
Thank you very much for the reviewer's comments. If there is anything that needs to be corrected, please inform us again and we will apply it right away.

Point 6: Would combining sections 2 and 3 into a single section with subsections be better?

Response 6: We thank you for your comment and appreciate informing us. We have corrected that combining sections 2 and 3 into a single section with subsections.
Thank you very much for the reviewer's comments. 

Point 7: In Figure 3, the legend indicates both energy and data flow. However, the configuration diagram does not have any data flow within it. Is this the case, or is it an error? Kindly check and update.
Response 7: We thank you for your comment and appreciate informing us. We modified the Figure 3 data line to be clearly visible. Please refer atteched note file.
Thank you very much for the reviewer's comments. 

Point 8: Figure 4 has different line styles. But there is no legend to interpret the different line styles. Hence, it is difficult to comprehend the diagram.
Response 8: We thank you for your comment and appreciate informing us. We added the the legend in the Figure 4. Please refer atteched note file.
Thank you very much for the reviewer's comments. If there is anything that needs to be corrected, please inform us again and we will apply it right away.

Point 9: Some critical elements of the paper are summarised in lines 205 to 212. These should be part of the introduction, except if there is a reasonable justification for the current position.

Response 9: We thank you for your comment and appreciate informing us. The contents (lines 205 to 212 in old version) pointed out by the reviewer was summarized and added to the previous introduction part.
Thank you very much for the reviewer's comments. If there is anything that needs to be corrected, please inform us again and we will apply it right away.

Reviewer 3 Report

The work introduces a future energy stabilization mechanism for peak power management in buildings and present a platform that entails connection-based energy trading technology based on a scenario

1. Please use any matrix to measure the energy stabilizing ability.

2. Please write an algorithm for the whole method.

3. Please compare your method with the state-of-the-art methods.

4. The contribution of the work is limited. The author can include results for other benefits of stabilizing the future grid system.

Author Response

Point 1: Please use any matrix to measure the energy stabilizing ability.

Response 1: We thank you for your comment and appreciate informing us.
Our proposed study is to analysed energy bills for controlling the peak energy in buildings through energy saving and energy supply technology. We really thought deeply about the metrics-based solution based on the reviewer's comment. We will also advance our research through a matrix-based mathematical approach. We believe that if we follow the reviewer's comments, the results of the study will be much better.
I would appreciate it if you could understand our circumstances.
Ask for reviewer's understanding.
Thank you very much for the reviewer's comments. 

Point 2: Please write an algorithm for the whole method.

Response 2: We thank you for your comment and appreciate informing us.
We have added the “System Connection Process and algorithm” in page 17~18 based on the reviewer's comments. And please refer atteched note file.
Thank you very much for the reviewer's comments. If there is anything that needs to be corrected, please inform us again and we will apply it right away.

Point 3: Please compare your method with the state-of-the-art methods.

Response 3: We thank you for your comment and appreciate informing us.
We have comprehensively modified Chapters “1. Introduction” and “2 Related Work” parts to represent the characteristics and difference of the proposed system.
Thank you very much for the reviewer's comments. If there is anything that needs to be corrected, please inform us again and we will apply it right away.
And please refer atteched note file.

Point 4: The contribution of the work is limited. The author can include results for other benefits of stabilizing the future grid system.

Response 4: We thank you for your comment and appreciate informing us.
We have modified Chapters “1. Introduction” and added the “Concept of Proposed PV + ESS, and FEETS Connection System” and “Difference of Existing System and Proposed System”.
Thank you very much for the reviewer's comments. If there is anything that needs to be corrected, please inform us again and we will apply it right away.

Reviewer 4 Report

The goal of this paper, as exposed by the authors, is to present an energy sharing scenario for energy stabilization assuming that electric vehicles and their charging stations are widely distributed in the future.

Related work is too short and concise. In order for the article to be accepted for publication, this part, which can be made as a separate chapter, must be significantly improved both in terms of content size and quality.

The aspects related to the network security and authentication must be explained, presented and validated with practical data.

In Fig 9.b the type of communication between server and EVi user (stage 1, stage 3 and stage 4) must be specified. Writing only ‘query’ and ‘response’ is not enough for a MDPI Buildings journal. A scientific article must also include technical information of the proposed concept implementation.

The three-step energy supply strategy algorithm and the proposed system scenarios are successful presented in the context of building energy peak. On which OSI level is the EVi – smart Gateway (Figure 10) implemented?

The authors should discuss the research gap and existing problems in the introduction section as the research motivation. Additional, the authors should summarize their main contributions in this study in bullets in the end of the Introduction section. For each point mentioned in the contribution paragraph, identify which part in the resubmitted manuscript considers that point. In particular, it is not clear whether the authors' contributions to THIS publication.

The reference section is good, citing new and relevant articles in the research area.

Author Response

Point 1: Related work is too short and concise. In order for the article to be accepted for publication, this part, which can be made as a separate chapter, must be significantly improved both in terms of content size and quality.

Response 1: We thank you for your comment and appreciate informing us.
We have added the “Related work chapter”, and we highlighted the differentiation of our research by adding various existing research and related work. Please refer the atteched file and thank you very much for the reviewer's comments for our paper.

Point 2: The aspects related to the network security and authentication must be explained, presented and validated with practical data.

Response 2: We thank you for your comment and appreciate informing us.
We have added the reviewer’s comment about the network security and authentication of the proposed system as below.

1) (Page 7~9) We added the chapter “3.1.2. Proposed System Architecture (Connection of CNDIP)”.  
-> We added the Security module for energy trading.
2) (Page 17~18) We added the chapter “5.4. System Connection Process based on the FEETS overall Scenario” -> We added the Security networking for energy trading.

Thank you very much for the reviewer's comments for our paper.

Point 3: In Fig 9.b the type of communication between server and EVi user (stage 1, stage 3 and stage 4) must be specified. Writing only ‘query’ and ‘response’ is not enough for a MDPI Buildings journal. A scientific article must also include technical information of the proposed concept implementation.

Response 3: We thank you for your comment and appreciate informing us.
We have added the chapter “5.4. System Connection Process based on the FEETS overall Scenario”. We specified the scenario and added the system connection process algorithm. Please refer the atteched file. Thank you very much for the reviewer's comments for our paper.

Point 4: The three-step energy supply strategy algorithm and the proposed system scenarios are successful presented in the context of building energy peak. On which OSI level is the EVi – smart Gateway (Figure 10) implemented?

Response 4: We thank you for your comment and appreciate informing us.
We consider that EV and gateway belong to the physical layer, and they have a structure that can link data through the data layer.
Thank you very much for the reviewer's comments for our paper. If there are anything that needs to be corrected, please inform us again and we will apply it right away.

Point 5: The authors should discuss the research gap and existing problems in the introduction section as the research motivation. 

Response 5: We thank you for your comment and appreciate informing us.
We have comprehensively modified Chapters “1. Introduction” and “2 Related Work” parts to represent the characteristics and difference of the proposed system.
Thank you very much for the reviewer's comments. If there is anything that needs to be corrected, please inform us again and we will apply it right away.
Please refer the atteched file.

Point 6: Additional, the authors should summarize their main contributions in this study in bullets in the end of the Introduction section. For each point mentioned in the contribution paragraph, identify which part in the resubmitted manuscript considers that point. 
In particular, it is not clear whether the authors' contributions to THIS publication.

Response 6: We thank you for your comment and appreciate informing us.
We have added contribution paragraph in the end of the Introduction section(page 4).
Thank you very much for the reviewer's comments. If there is anything that needs to be corrected, please inform us again and we will apply it right away.
Please refer the atteched file.

Point 7: The reference section is good, citing new and relevant articles in the research area.

Response 7: We thank you for your comment and appreciate informing us. We are very grateful for the reviewer's evaluation of the paper.

Reviewer 5 Report

1. discuss advantages of research in the end of abstract

2. cite figure and tables if they are related to literature reveiew

3. justify all methodology by citing the references, please dont write blind

Author Response

Point 1: discuss advantages of research in the end of abstract.

Response 1: We thank you for your comment and appreciate informing us.
We have added the advantages of research in the end of abstract.
Thank you very much for the reviewer's comments for our paper.
Please refer the atteched file.

Point 2: cite figure and tables if they are related to literature reveiew

Response 2: We thank you for your comment and appreciate informing us.
We have corrected the literature review and we have added the “Related Work” chapter in the paper. 
Thank you very much for the reviewer's comments. If there is anything that needs to be corrected, please inform us again and we will apply it right away.
Please refer the atteched file.

Point 3: justify all methodology by citing the references, please dont write blind

Response 3: We thank you for your comment and appreciate informing us.
We tried to advance the paper by examining the references according to the comments of the reviewer. We cited past our related papers and added references by researching other relevant papers.
Thank you very much for the reviewer's comments. If there is anything that needs to be corrected, please inform us again and we will apply it right away.

Round 2

Reviewer 3 Report

I think Authors has addressed all the comments I was concerned.

Reviewer 4 Report

I think that the paper can be accepted.